# A microfluidic platform integrating functional vascularized organoids-on-chip

Clément Quintard[1,2,3], Emily Tubbs[1], Gustav Jonsson [4,5,6], Jie Jiao[3], Jun Wang [3], Nicolas Werschler [7], Camille Laporte [1,2], Amandine Pitaval[1], Thierno-Sidy Bah[8], Gideon Pomeranz[9], Caroline Bissardon[2], Joris Kaal[2], Alexandra Leopoldi[4,6], David A. Long [9], Pierre Blandin[2], Jean-Luc Achard[10], Christophe Battail [8], Astrid Hagelkruys [4,6], Fabrice Navarro[2], Yves Fouillet[2], Josef M. Penninger [3,4,6,11] ✉ & Xavier Gidrol [1] ✉

The development of vascular networks in microfluidic chips is crucial for the long-term culture of three-dimensional cell aggregates such as spheroids, organoids, tumoroids, or tissue explants. Despite rapid advancement in microvascular network systems and organoid technologies, vascularizing organoids-on-chips remains a challenge in tissue engineering. Most existing microfluidic devices poorly reflect the complexity of in vivo flows and require complex technical set-ups. Considering these constraints, we develop a platform to establish and monitor the formation of endothelial networks around mesenchymal and pancreatic islet spheroids, as well as blood vessel organoids generated from pluripotent stem cells, cultured for up to 30 days on-chip. We show that these networks establish functional connections with the endothelium-rich spheroids and vascular organoids, as they successfully provide intravascular perfusion to these structures. We find that organoid growth, maturation, and function are enhanced when cultured on-chip using our vascularization method. This microphysiological system represents a viable organ-on-chip model to vascularize diverse biological 3D tissues and sets the stage to establish organoid perfusions using advanced microfluidics.

The ability to vascularize organoids remains a challenge in the field of tissue engineering. Indeed, most tissues exceeding 400 μm in thickness need a functional vasculature to ensure a sufficient supply of nutrients and oxygen, as well as the ability to remove carbon dioxide and cellular waste products, preventing the formation of necrotic inner cores[1]. Although significant efforts have been directed towards creating increasingly complex organoid model systems in vitro, it remains necessary to transplant such organoids into host animals to establish functional vascular circulation[2–6]. However, in vivo transplantations are very expensive and lack scalability for larger scale toxicity or drug screening.

[1]Univ. Grenoble Alpes, CEA, IRIG/BGE, BIOMICS, 38000 Grenoble, France. [2]Univ. Grenoble Alpes, CEA, LETI, DTBS, 38000 Grenoble, France. [3]Department of Medical Genetics, Life Sciences Institute, University of British Columbia, Vancouver, British Columbia (BC), Canada. [4]Institute of Molecular Biotechnology of the Austrian Academy of Sciences, IMBA, Dr. Bohr-Gasse 3, 1030 Vienna, Austria. [5]Vienna BioCenter PhD Program, Doctoral School of the University of Vienna and Medical University of Vienna, 1030 Vienna, Austria. [6]Eric Kandel Institute, Department of Laboratory Medicine, Medical University of Vienna, Vienna, Austria. [7]School of Biomedical Engineering, University of British Columbia, Vancouver, British Columbia (BC), Canada. [8]Univ. Grenoble Alpes, CEA, IRIG, BGE, Gen&Chem, 38000 Grenoble, France. [9]Developmental Biology and Cancer Programme, UCL Great Ormond Street Institute of Child Health, London, UK. [10]Université Grenoble Alpes, CNRS, Grenoble INP, LEGI, 38000 Grenoble, France. [11]Helmholtz Centre for Infection Research, Braunschweig, Germany. ✉e-mail: josef.penninger@ubc.ca; xavier.gidrol@cea.fr

Considerable effort has been directed towards addressing this problem through the generation of perfusable vascular networks on-chip, either as standalone[7–9], or by incorporation of other tissues to develop functional, vascularized organs-on-chips[10–15]. In this in vitro approach, endothelial cells and supportive cells are seeded into a central microfluidic chamber using hydrogels. These hydrogels provide structural support to the embedded cells where they can self-organize into endothelial networks. The media flows continuously into the lateral channels adjacent to the central microchamber, providing the nutrients and gas exchange required for long-term cell culture. However, this conventional geometry does not allow reproduction of the fluxes observed in vivo, and replicating the in vivo functional vascularization of iPSC-derived organoids is still an ongoing challenge[16].

Here, we report a platform to vascularize various biological tissues on-chip, using an original and user-friendly microfluidic device and chip loading process. The reliability of our system was validated using spheroids generated from human fibroblasts and endothelial cells as well as 3D human blood vessel organoids (BVOs) generated from human-induced pluripotent stem cells (hiPSCs)[3,17] and human pancreatic islet spheroids. Importantly, we demonstrate effective anastomosis and controlled perfusion of the vascular organoids, as well as enhanced organoid growth, maturation, and vasculature development. Additionally, we report enhanced functionality of pancreatic islet spheroids using our platform.

## Results

### Design of a microfluidic device for precise encapsulation of organoids

We set out to design a microfluidic device that offers reliable and user-friendly monitoring of flow dynamics. A microfluidic chip was fabricated using cyclic olefin copolymer (COC), a material that presents long-term robustness, is suitable for mass production, exhibits desired optical qualities for imaging, and low absorption of chemicals[18]. Each chip contained 10 microchannels and was monitored using a 10-channel syringe-pump (Fig. 1a–c and Supplementary Fig. 1). We next developed an efficient method, adapted from existing hydrodynamic trapping principles[19], that allowed for the encapsulation of organoids in a predefined location within a serpentine-shaped microchannel (Fig. 1d, Supplementary Fig. 2a and Supplementary Note 1)[20]. The organoids, embedded in a fibrin hydrogel, were indeed precisely positioned at the trap site without any apparent morphological alteration (Supplementary Fig. 2b). Of note, the dimensions of the trap site can be adjusted according to the size of the organoids used (for BVOs of diameter $\emptyset \approx 600\,\mu m$, Width = 300 μm and Height = 800 μm, for mesenchymal and pancreatic islet spheroids of diameter $\emptyset \approx 300\,\mu m$, Width = 200 μm and Height = 400 μm) (Fig. 1d and Supplementary Fig. 1d).

Into the fibrin hydrogel we incorporated a mixture of human umbilical vein endothelial cells (HUVECs) and fibroblasts, surrounding the organoid. Subsequently, air was injected to push the hydrogel toward the exit of the microfluidic channel. After allowing the gel to polymerize for approximately 5 minutes at room temperature, continuous microfluidic perfusion with growth medium was established (Fig. 1e). Both the hydrogel and air were injected at $Q = 300\,\mu l/min$. Of note, the hydrogel remained inside the trap site due to capillarity, effectively surrounding the organoid and thereby minimizing organoid contact with the microchannel walls resulting in a permanent organoid encapsulation inside the trap site[20]. Because of the Landau-Levich-Bretherton effect[21], a thin layer of hydrogel remained along the square profile cross-section of the microchannels after injection of air (Fig. 1f). We used this property as means of achieving complete endothelialization of the serpentine channel with HUVECs (Supplementary Fig. 3a). Experiments using an adapted in-house light sheet fluorescence microscopy set-up were conducted to confirm the

three-dimensional structure of the hydrogel deposition near the trap site and along the main serpentine microchannel (Fig. 1g and Supplementary Fig. 3b–d). The thickness of the gel layer on the microchannel walls can be adjusted by varying the flow rate of the air bubble passage (Supplementary Fig. 3e). This localized hydrogel deposition enabled the development of a self-organized endothelial network that traverses the trap site and supplies the organoid with nutrients. One of the strengths of our microfluidic platform is its robustness, scalability, and adaptability. Whereas other platforms might achieve comparable vascularization, the presented design promises enhanced speed and reliability. In practice, we can load a microchannel in just around 10 seconds, achieving a trapping efficiency that is close to 100% (Supplementary Fig. 4).

Moreover, this chip design can be readily modified to host several traps in order to study multiple organoids in parallel (Supplementary Fig. 5). In this study, our emphasis was on vascular spheroids and organoids (generated off-chip) cultured in a fibrin hydrogel. However, our approach is versatile and applicable to other commonly utilized extracellular matrices (ECM) like Matrigel. For instance, by adhering to the same protocol, we cultured hiPSC-derived lung organoids within Matrigel (Fig. 1h). These organoids were not only efficiently trapped in our device but also exhibited robust growth and bud formation over a 2-week culture period on-chip. Overall, we have designed a serpentine geometry chip for precise and controlled entrapment of organoids and other 3-dimensional cellular constructs in endothelium-lined microfluidic channels.

### Establishment of interconnected perfusable endothelial networks

To interrogate the utility and biological relevance of our system, we used cell aggregates consisting of human fibroblasts and GFP labelled HUVEC cells, termed hereafter mesenchymal spheroids, which were seeded into the microfluidic channels. We first examined the effects of fluid flow by culturing the mesenchymal spheroids alone under static (media changed daily) or flow conditions (Fig. 2a). We observed an enhanced formation of endothelial networks under dynamic perfusion, with a significant 4.4-, 6.5-, 5.0- and 4.8-fold increase in the number of vessel junctions, number of meshes, number of segments and total segment length, respectively, as compared to static conditions. Thus, flow conditions can directly drive the differentiation of these mesenchymal spheroid into vessel-like structures (Fig. 2a–c).

We next incorporated HUVECs in the hydrogel mix (as described above) to establish functional connections between the HUVEC lined microchannel and the trapped mesenchymal spheroid. The HUVEC endothelial cells of the spheroid expressed GFP and the HUVEC endothelial cells suspended inside the gel prior to injection expressed RFP, allowing for visualization of the distinct cell populations and their interactions in real time (Fig. 2d). On day 0, the mesenchymal spheroid was introduced into the microchannel and trapped at the correct location, where it maintained its spherical shape. On day 3, we observed an initial organization of the endothelial cells, and by day 7, network-like structures with a three-dimensional configuration developed and remained stable up to day 13 (Fig. 2d). Intriguingly, we observed spontaneous anastomosis between the RFP-HUVEC endothelial bed and the vasculature of the mesenchymal spheroids (Fig. 2d inset), establishing an interconnected network across the trap site.

To demonstrate the functionality of the interconnected endothelial networks, we performed perfusion assays using fluorescent microbeads (Fig. 2e). To visualize flow through the network, beads with a diameter of 1 μm were injected into the microchannel at a flow rate of $Q_+ = 10\,\mu l/min$ on the day 13 of culture. The bead tracings were overlaid onto the patterns generated by the endothelial networks from the entrance to the exit of the trap, indicating the capability of the formed network to support perfusion (Fig. 2f, Supplementary Fig. 6a and Supplementary Movie 1). Importantly, the majority of the beads

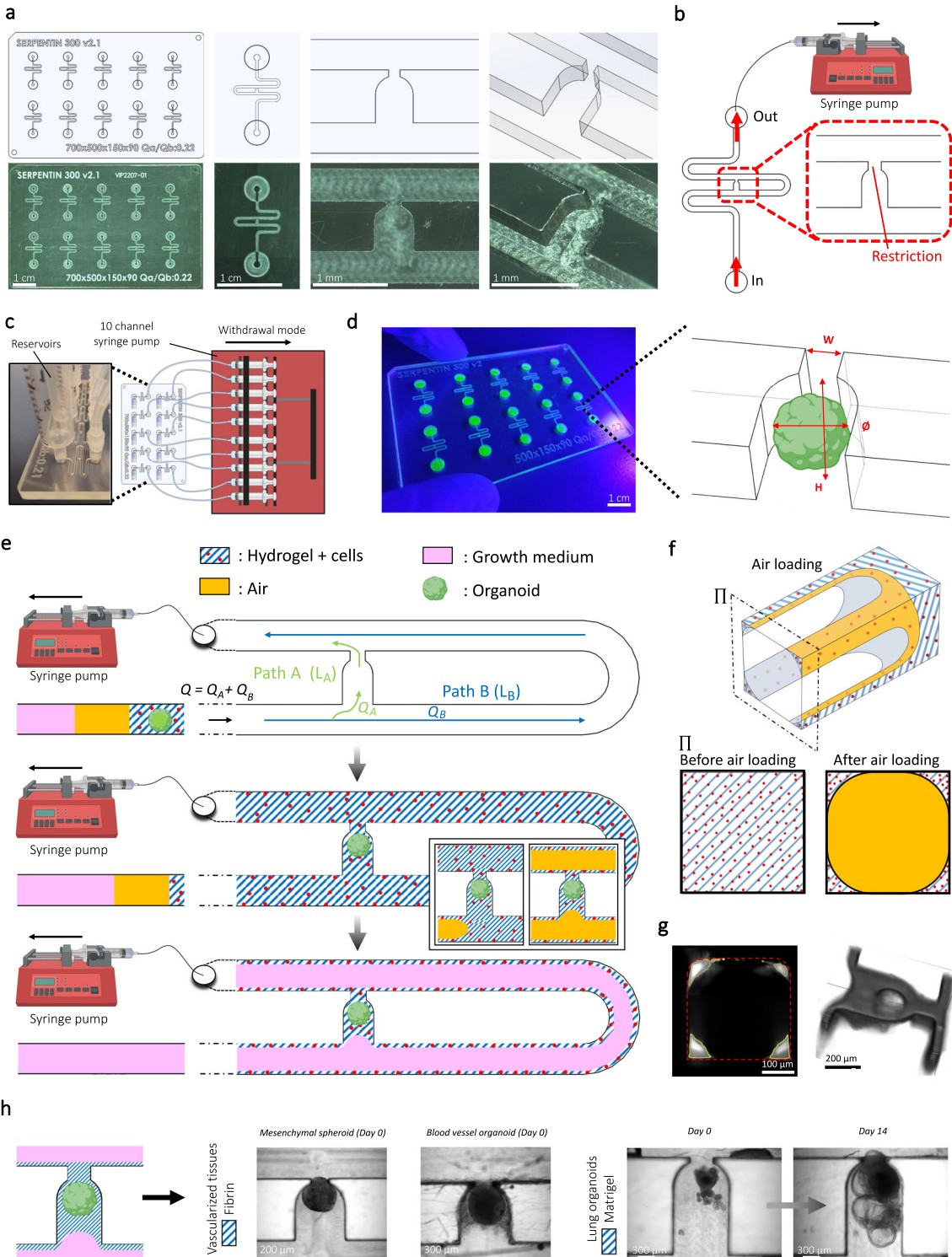

**Fig. 1 | Device design and overview of organoid and cell configurations.**
**a** Computer-aided design (top) and photographs (bottom) of the microfluidic chip displaying 10 microchannels, with each microchannel featuring a trapping site. **b** Top view of the microfluidic device. A syringe pump was connected to the outlet of the channel to introduce fluid perfusion. **c** Schematic diagram and photograph of the parallelization feature of our setup, showcasing 10 microchannels controlled simultaneously. **d** Photograph of the microfluidic chip and schematic three-dimensional view of the U-cup shaped area functioning as a trap. Here, the trap site is exemplarily occupied by a cell aggregate. **e** Schematic diagram showing an overview of the loading process. Initially, the hydrogel containing an organoid and HUVEC cells was introduced. Before polymerization of the hydrogel, air was

introduced to position the hydrogel and the HUVEC cells. Finally, growth medium was introduced for continuous perfusion of the microfluidic chamber and the trapped organoid. **f** Schematic 3D and cross-sectional views of the microchannel showing the air loading process and associated hydrogel deposition.
**g** Experimental cross-sectional view (left) of the microfluidic channel showing the hydrogel deposition in the trap and in the channel's corners and 3D rendering (right), taken with an in-house light sheet fluorescence microscopy set-up.
**h** Representative images of vascular spheroid/organoid cultured in fibrin (left), and hiPSC-derived lung organoids cultured in Matrigel, showing efficient trapping and robust growth over 2 weeks on-chip (right).

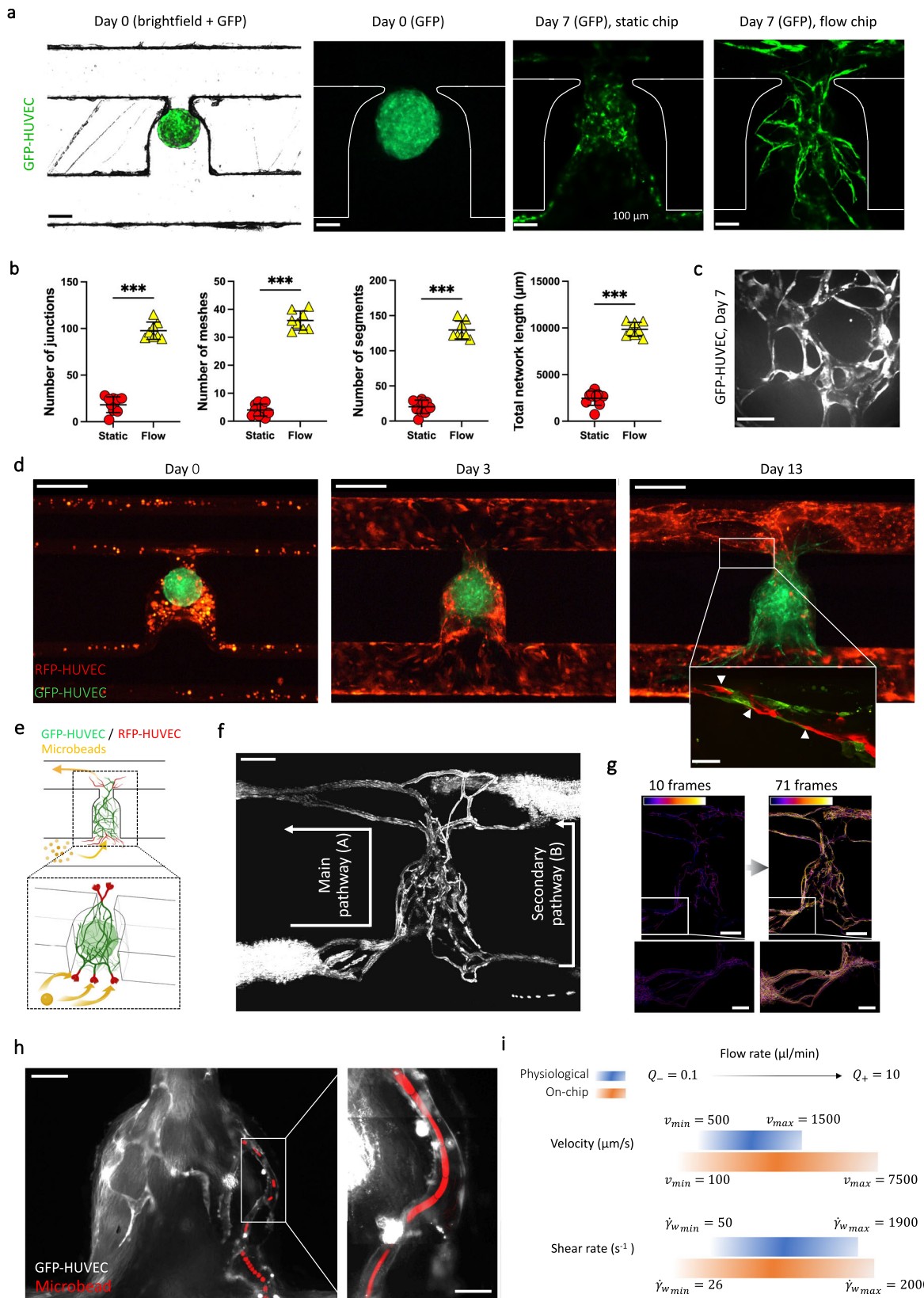

travelled through the endothelial network along the main perfusion direction (path A); as expected, some beads also flowed through the secondary pathway (path B) (Fig. 1d, f). Of note, as the experiment was done at a constant flow rate, the velocity of a microbead was inversely proportional to the cross-sectional area it flowed through, hence the velocity decreased in regions without endothelium. This resulted in an

intense fluorescent signal from the beads' tracings on the bottom left and top right corners of the imaged area (Fig. 2f).

We next analyzed the distribution of the microbeads within the endothelial network over time; this was done by counting the numbers of pixels in each frame of the binarized and frame-color-coded movies that corresponded to the movements of the microbeads. The

**Fig. 2 | Generation of anastomosed endothelial networks through functional vascularization of mesenchymal spheroids. a** Mesenchymal spheroids cultured on-chip under static or flow conditions. Representative images are shown at day 7 after seeding. **b** Angiogenesis Analyzer output of the morphology of the endothelial networks after a week of culture on-chip ($n = 10$ (static) and 8 (flow)). **c** Confocal z-stack maximum intensity projection of the three-dimensional endothelial network. **d** Gel embedded RFP-HUVEC cells in the main channel are shown in red, GFP-HUVEC cells from the mesenchymal spheroids are shown in green. Note the formation of structured endothelial networks over time that appear to be stable until the end of the observation period (day 13 after seeding). In a close-up view, anastomosis between RFP-HUVEC and GFP-HUVEC cells are indicated by white arrows (inset). **e** Schematic drawing depicting the cell culture configuration, with anticipated open connections at the fibrin gel interface allowing for microbeads perfusion (created with BioRender.com). **f** Maximum of intensity projection over a 71 images stack, highlighting the tracks of the microbeads passing through the interconnected network. See Supplementary Movie 1 and Supplementary Fig. 5 for raw movies and details. **g** Sum of the binarized and frame-color coded images from (**f**) showing time-resolved beads perfusion. See Supplementary Movie 2 for details. **h** Projections of maximum intensity over an image stack showing tracking of one individual microbead (red) passing through the endothelial network. The inset shows an assembled projection of three movies taken at the indicated area at higher magnification. For better visualization, the RFP-HUVEC cells are not shown in these images. See Supplementary Movie 3 for raw data. **i** Summary of different flow parameters measured in our organ-on-chip device as compared to in vivo physiological flow rates in human capillaries. Data represent mean values ± s.d. Beads were 1 μm (**f**), 4.8 μm and 0.5 μm (**h** and inset) in diameter. Scale bars, 400 μm (**d**), 200 μm (**f** and **g**), 100 μm (**a**, **c** and **h**), 50 μm (**d** (inset)) and 20 μm (**h** (inset)). Data represents mean ± s.d. Statistical significance was attributed to values of $P < 0.05$ as determined by unpaired t test (two-tailed). ***$P < 0.001$. Source data are provided as a Source Data file.

microbeads perfused the whole endothelial network without any apparent prioritization of any particular area (Fig. 2g, Supplementary Fig. 6b and Supplementary Movie 2), demonstrating that apparently the entire endothelial network present in the mesenchymal spheroids was perfusable. In the microchannels shown in Fig. 2h, a smaller number of beads was introduced at a lower flow rate of $Q_- = 0.1\,\mu l/min$ to visualize the motions of individual microbeads (Supplementary Movie 3). Superposition of the tracked microbeads (red) and the fluorescent signal from the mesenchymal cell aggregate endothelium (grey) confirmed perfusion of the network (Fig. 2h). Thus, our serpentine geometry chips enable the establishment of anastomosed and perfusable endothelial networks.

Lastly, we investigated the physiological relevance of our microfluidic platform. The observed flows in the recorded networks were characterized by tracking fluorescent microbeads as they entered the endothelial network. Depending on the flow rate imposed by the syringe pump, microbeads moving through the microvessels exhibited fluid velocities ranging from $v_{min} = 100\,\mu m/s$ to $v_{max} = 7500\,\mu m/s$ (Fig. 2i). With a calculated Reynolds number (Re) of $10^{-3}$ to $10^{-1}$ (Supplementary Note 2), the flow can be ascertained to be laminar; thus one can deduct from these values the fluidic shear rate at the vessel wall, which is given by $\dot{\gamma}_w = \frac{4\bar{v}}{R}$, where $R$ is the radius of the vessel and $\bar{v}$ the linear fluid velocity (Supplementary Note 3). Using this equation, $\dot{\gamma}_w$ ranged from 27 to 2000 $s^{-1}$. The mean (± s.d.) velocity over the perfused network at $Q_-$ and associated shear rate, were $\bar{v} = 502 \pm 305\,\mu m/s$ and $\dot{\gamma}_w = 134 \pm 81\,s^{-1}$, respectively ($n = 4$ beads individually tracked). The mean (±s.d.) velocity over the perfused network at $Q_+$, and the associated wall shear rate, were $\bar{v} = 2689 \pm 531\,\mu m/s$ and $\dot{\gamma}_w = 531 \pm 142\,s^{-1}$, respectively ($n = 4$ beads individually tracked). The observed range of flux values was consistent with the flow rates detected in human blood vessels of similar size[22], corresponding to velocities of $\bar{v} = 500–1500\,\mu m/s$ and wall shear rates of $\dot{\gamma}_w = 50–1900\,s^{-1}$ (Fig. 2i). Thus, the observed perfusion flow in our organ-on-chip device resembles physiologic flow observed in human capillaries.

## Vascularization of blood vessel organoids

While significant progress has been made in the development of complex organoids, the convergence of human tissue engineering and microfluidics is needed to address the technical challenges that remain, in particular to vascularize 3D tissues to support extended growth and differentiation[23]. To further show the utility of our device design, we seeded 3D blood vessel organoids (BVOs) generated from human-induced pluripotent stem cells (hiPSCs) into our chip. We have previously reported the generation of such organoids[17] and demonstrated their physiological relevance through the modelling of diabetic vasculopathy both in vitro and in vivo[3]. Slight modifications were made to the BVO differentiation protocol to ensure diametric homogeneity for trapping consistency (Fig. 3a). Similarly to BVOs cultured in wells, these BVOs self-organize on-chip into three-dimensional interconnected networks of bona fide capillaries containing an endothelial cell lined lumen, a pericyte and smooth muscle cells coverage, and expression of tight junction proteins ZO-1 and adherens junction protein VE-cadherin (Fig. 3b). The organoids formed capillaries with hollow lumens surrounded by a prototypic basal membrane (Fig. 3c). Because it has been previously reported that narrow capillaries fail to open at the interface to the media channel[24], we attempted to link these BVO capillaries with the HUVEC network of the microchannel.

Mature BVOs (day 15), GFP-HUVECs ($6 \times 10^6$ cells/ml) and fibroblasts ($2 \times 10^6$ cells/ml) were embedded within the hydrogel and introduced into the microchannels as described above (Fig. 1d). HUVECs self-organized into endothelial networks arborizing the seeded BVOs and were cultured under long-term continuous perfusion on-chip for up to two weeks (Fig. 3d). HUVEC networks with hollow lumen structures were observed starting around one week of culture (Fig. 3e). Starting from a single-cell suspension, HUVECs begin to self-organize within the first day of culture. By day 6, we observed the establishment of stable endothelial networks surrounding the BVOs, with open connections to the main microchannel (indicated by white arrows), which remained from that day onwards (Fig. 3f). Using the Angiogenesis Analyzer plugin in ImageJ, the development of the vascular network was quantified (Fig. 3g and Supplementary Fig. 7). We measured an average increase of 110%, 109% and 96% between day 1 and day 10 in the total network length (Fig. 3h), numbers of segments (Fig. 3i) and numbers of junctions (Fig. 3j), respectively. Moreover, between day 2 and day 10, we observed an average decrease of 44% in the total isolated branches length (Fig. 3k), indicative of a progressive organization into an interconnected network.

The networks formed endothelial beds encompassing the BVOs and exhibited three-dimensional organization that extended across the trap site (Fig. 4a). After 10 to 14 days of on-chip culture, these cellular structures were fixed and stained using a dynamic on-chip protocol where PFA 4%, blocking buffer, primary antibody and secondary antibody solutions were sequentially flowed into the microchannels. All the GFP-HUVEC vessels were CD31 positive (CD31+), showing that the microvascular networks can be readily perfused with the anti-CD31 antibody (Fig. 4b). Functionality of the HUVEC networks was assessed by the live perfusion of fluorescent microbeads through the microchannels after 10 to 13 days of culture (Fig. 4c and Supplementary Movie 4). Note that the beads were nearly completely flushed out during the immunostaining on-chip process, however, a few beads remained trapped inside the HUVEC formed endothelial structures (Fig. 4d). While most of the experiments were stopped after around 2 weeks of culture on-chip, the HUVEC networks were found to be stable up to 30 days of culture (Fig. 4e). These findings demonstrate the robustness of our microfluidic platform to form stable and perfusable endothelial networks, arborizing the trapped BVOs.

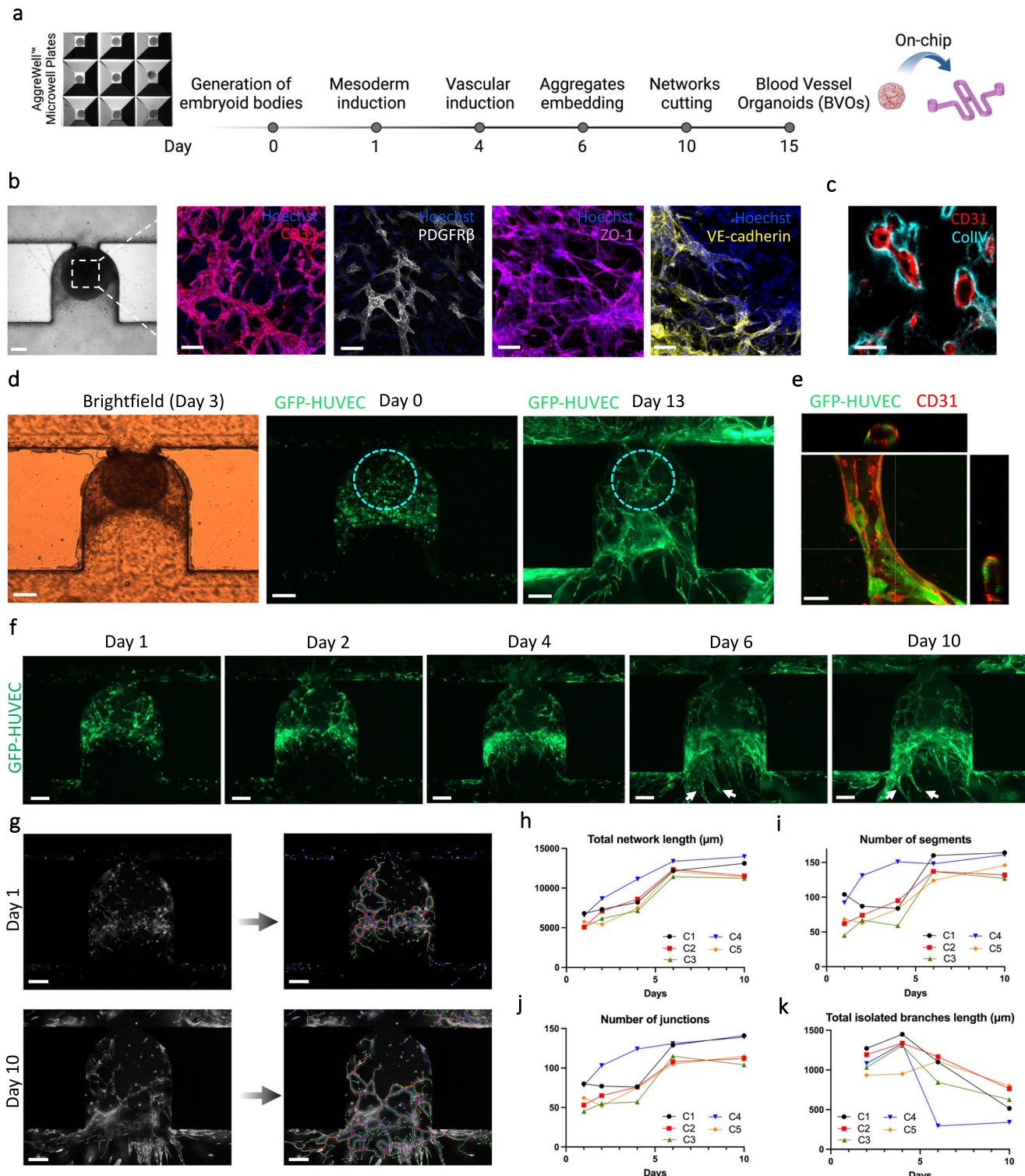

**Fig. 3 | Vascularization of Blood Vessel Organoids (BVOs) on-chip. a** Schematic of the protocol used for the differentiation of human pluripotent stem cells into blood vessel organoids. **b** Representative immunofluorescence of BVOs-on-chip, with capillary networks expressing CD31, the tight junction protein ZO-1, the adherens junction protein VE-cadherin, and covered by pericytes (PDGFRβ). **c** Cross-section of BVO-on-chip showing capillaries with hollow lumens and collagen type IV basal membrane coverage. **d** Initial and final cell culture configuration. **e** Vessel from the endothelial HUVEC network (stained for the endothelial marker CD31) with orthogonal views showing hollow lumen structures. **f** Time resolved evolution of the cell culture on chip. GFP-HUVEC cells self-organized from a single cell suspension, into an endothelial network surrounding the organoid. **g** Quantification of the endothelial networks using Angiogenesis Analyzer plugin, at different time points of the on-chip culture. **h**–**k** Angiogenesis Analyzer outputs of the morphology of the endothelial networks regarding the total network length (**e**), the number of segments (**f**), the number of junctions (**g**) and the total isolated branches length (**h**) in imaged area from $n = 5$ independent microchannels denoted as $C_i$ (where i ranges from 1 to 5). Scale bars, 200 μm (**b**, **d**, **f**, and **g**) and 20 μm (**c**, **e**).

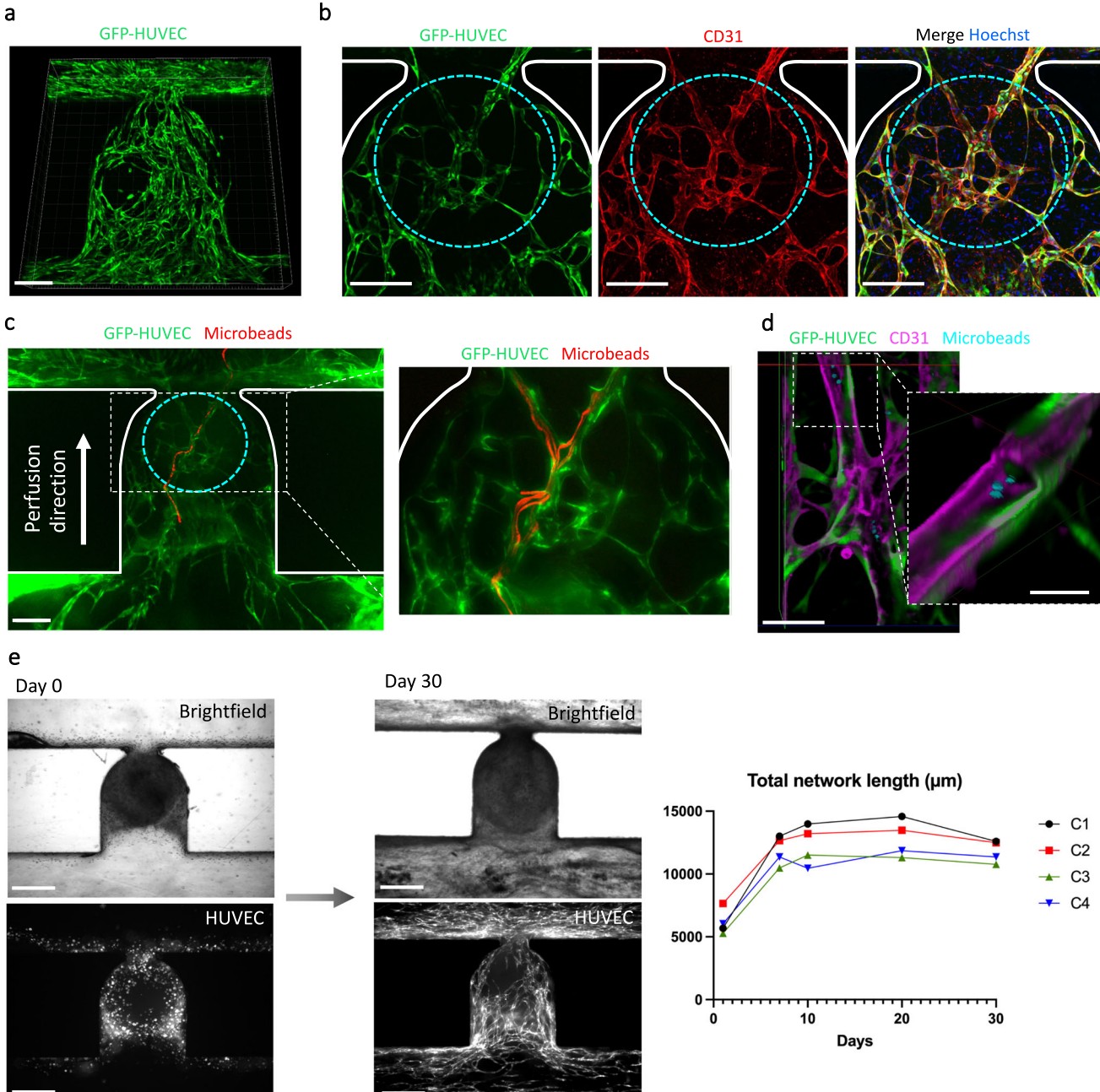

**Fig. 4 | Establishment of perfusable HUVEC endothelial networks encompassing blood vessel organoids. a** Imaris 3D rendering from confocal z-stacks of the endothelial network that has developed in the trapping site after 14 days of culture on-chip. **b** Confocal z-stack maximum intensity projection of the endothelial network at day 14 after staining of the microfluidic chip for CD31 expression and the nuclear marker Hoechst. Cyan dotted lines are used to provide the organoid's location behind the network, determined through brightfield images. **c** Projection of maximum intensity over an image stack showing the tracking of one individual microbead (red) passing through the endothelial network (green). The inset shows the perfusion of several microbeads at higher magnification. See Supplementary Movie 4 for raw movies. **d** 3D representation from confocal z-stacks using clipping planes to reveal the presence of fluorescent microbeads located inside the lumen of a HUVEC vessel. **e** Angiogenesis Analyzer outputs of endothelial networks on-chip assessing the total network length evolution over a period of 30 days. Experiments were conducted on $n = 4$ independent microchannels denoted as Ci (where i ranges from 1 to 4). Scale bars, 400 μm, (**e**), 200 μm (**a**–**c**), 50 μm (**d**) and 20 μm (**d** (inset)).

## Functional anastomosis between HUVEC networks and BVOs-on-chip

To demonstrate "functional vascularization", we examined anastomosis between the HUVEC endothelial networks and the seeded BVOs. Having previously shown that BVOs transplanted into the kidney capsule of mice formed functional connections with the host vasculature[3], we tested whether we could achieve similar in vitro anastomoses through the use of an intermediate HUVEC endothelial bed. On-chip anti-CD31 antibody staining revealed the presence of endothelial cells in the BVO vasculature central to the GFP-labelled HUVEC network (Fig. 5a). Numerous BVOs vessels (CD31⁺GFP⁻) were observed near HUVEC networks, suggesting functional anastomosis from the HUVEC endothelial bed into the organoid vessels (Supplementary Fig. 8a). When cultured under flow conditions, the vascularized BVOs displayed a physiological hierarchical organization with larger HUVEC vessels upstream and downstream of the BVO (Fig. 5b).

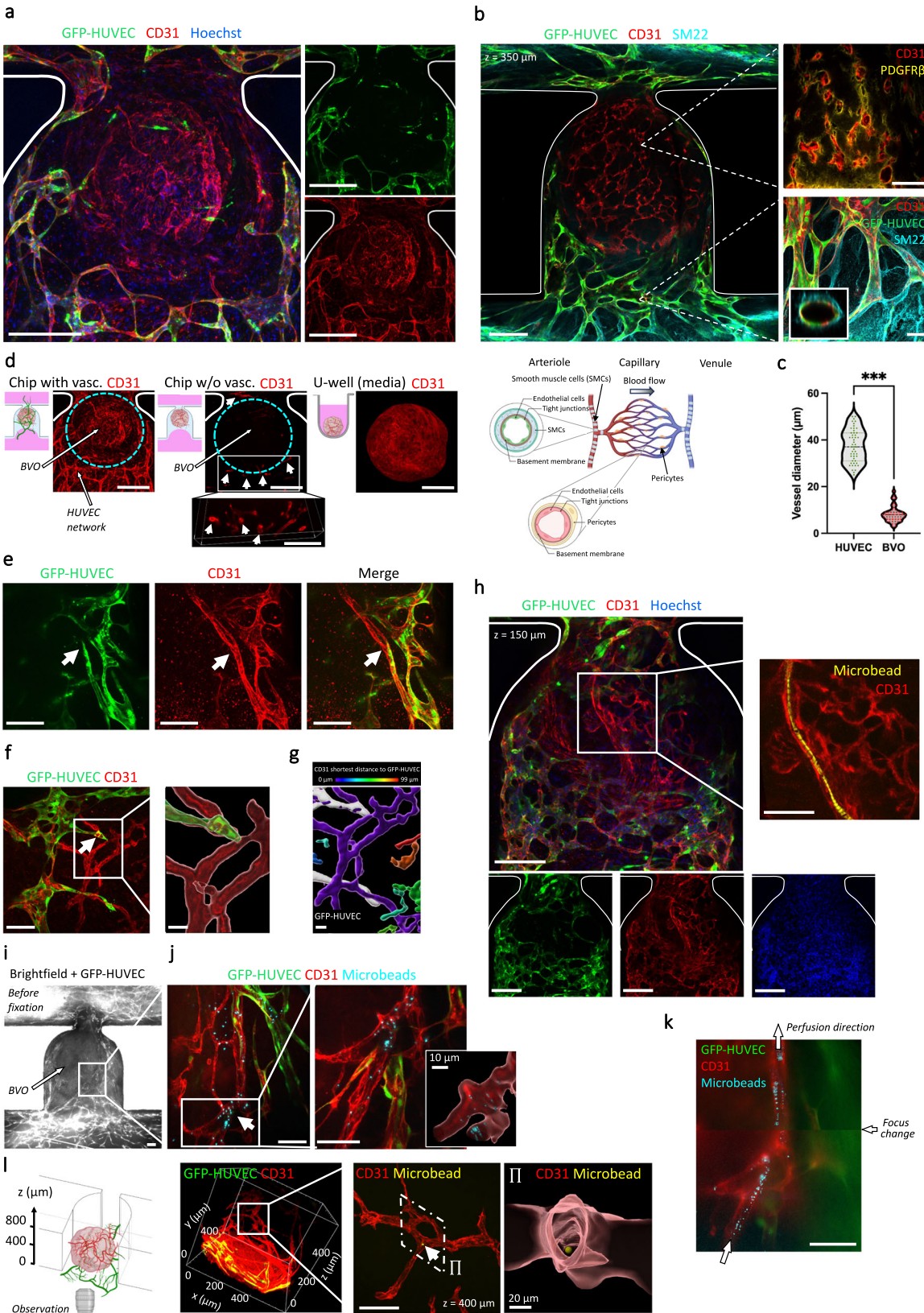

The regions upstream and downstream with arteriole-venule sized vessels exhibited strong expression of the smooth muscle cell SM22 marker, while the narrow capillaries in the BVOs had hollow lumen structures surrounded by pericytes (Fig. 5b). The average diameter of the HUVEC vessels in the areas upstream and downstream of the BVO was measured to be 37 μm, corresponding to arterioles-venules sizes, whereas the vessels inside the BVO averaged 8 μm in diameter, aligning with in vivo capillary sizes (Fig. 5c). Of note, BVOs that were cultured with fibroblasts but no HUVECs (referred to as "w/o vasc."), exhibited minimal CD31 staining and only few vessels at the periphery of the BVO, adjacent to the microchannel, were stained (Fig. 5d); however, we did detect endothelial sprouts with hollow lumens in the fibrin

**Fig. 5 | Assessment of functional anastomosis between a HUVEC endothelial bed and stem cell derived blood vessel organoids. a** Confocal z-stacks maximum intensity projection after staining of the microchannels for CD31 expression (red) and the nuclear marker Hoechst. The BVO vasculature corresponding to CD31$^+$GFP$^-$ vessels is visible in the center of the trap. **b** Hierarchical structure of vascularized BVO-on-chip emulates the arteriole-capillary-venule transition as seen in vivo. Schematic representation below (created with BioRender.com). **c** Quantification of diameters for HUVEC vessels and BVOs vessels (*n* = 50 individual vessels were measured for each type). **d** Confocal z-stacks maximum intensity projection after staining for CD31 expression of the organoid cultures under flow conditions with and without HUVECs, and in wells. The organoid location is shown by the cyan dotted line. **e–g** Anastomosis of GFP$^+$ HUVEC vessels with iPS cell-derived blood vessels of the organoid (**e**). Imaris segmentation and distance mapping revealed direct contact between the two vasculatures (**f** and **g**). **h** Z-slice (150 μm deep) from a confocal z-stack showing the tracking of one individual microbead (yellow) passing through the BVO endothelium. Side panels show red, green and blue color channels separately. **i, j** Vascularized BVO imaged live (**i**) and stained (**j**) after overnight beads perfusion. Microbeads accumulation in the BVO's vasculature is highlighted by a white arrow (**j**) and Imaris 3D rendering (**j**, inset). **k** Projection of maximum intensity over an image stack showing tracking of microbeads (cyan) passing through the BVO's vasculature (red). The perfusion direction is indicated by arrows and the image is an assembly from two movies taken at different z positions. See Supplementary Movie 5 for raw data. **l** Schematic diagram of the imaging setup (created with BioRender.com) and 3D rendering of a resulting confocal z-stack are shown. A close-up view of an individual microbead lodged deep within the BVO's vasculature is presented, approximately 400 μm away from the bottom of the microfluidic chip. Beads were 2 μm (**h**) and 1 μm (**j, k** and **l**) in diameter. Scale bars, 200 μm (**a, b** and **h**), 100 μm (**d–f** and **h** (inset), **i, k** and **l**), 30 μm (**f** (inset) and **g**). Statistical significance was attributed to values of *P* < 0.05 as determined by unpaired t test (two-tailed). ***P* < 0.001. Source data are provided as a Source Data file.

hydrogel in the "w/o vasc." condition (Fig. 5d inset). This data indicates that the presence of the intermediate endothelial bed is required for the antibodies to penetrate the BVOs located at the trapping area.

Importantly, we were also able to identify several interfaces showing anastomosis between the HUVEC endothelial network and the BVO's vasculature (Fig. 5e and Supplementary Fig. 8). Using segmentation tools in Imaris on the confocal fine z-stacks, we confirmed direct connections between HUVEC GFP$^+$ networks and BVO's CD31$^+$GFP$^-$ capillaries (Fig. 5f). Following distance transformation mapping, the organoid vasculature was further shown to be in direct contact (purple) with the HUVEC network (white) (Fig. 5g). We also performed microbeads perfusions on-chip. Considering fundamentals in fluid mechanics, the beads preferentially flow along the paths of least fluid resistance; therefore, the likelihood that beads will enter the narrow, lumenized vasculature of the organoid is low as a result of increased resistance, elevated intralumenal pressure, and reduction in flow-through. Nevertheless, we were able to capture microbeads within large vessels of the organoids in real time (Fig. 5h) and we were able to observe beads inside the internal vasculature of the BVOs after overnight perfusion (Fig. 5i, j). We also captured bead perfusion footage in a subset of stained organoids (Fig. 5k and Supplementary Movie 5). Moreover, we were able to image individual microbeads deep inside the BVOs' vasculature (Fig. 5l and Supplementary Fig. 9a, b), supporting the notion of effective organoid perfusion. Imaris segmentation confirmed the presence of beads inside the blood vessels (Fig. 5l inset). In the absence of HUVEC endothelial networks, the BVO's vasculature was not perfused as assessed by live microbeads perfusion, and no beads were detected in the trap site area (Supplementary Fig. 9c). Together, these results show that our method is suitable for connecting HUVEC endothelial networks with bona fide blood vessel organoids to organize a perfusable functional vascular tree.

## Enhanced on-chip blood vessel organoids growth and maturation

To demonstrate the applicability of our approach to address critical questions in the field concerning improved growth and functionality of 3D tissues via vascularization, we conducted experiments comparing organoid growth under both static and flow conditions, and with or without a neighbouring HUVEC endothelial bed (termed "with vasc." and "w/o vasc.", respectively). Images in brightfield and GFP fluorescence were taken repeatedly to observe the development of both the BVOs (circled in dotted cyan line for ease of visualization) and the surrounding endothelial networks (Fig. 6a, b). BVOs cultured on-chip in the fibrin hydrogel without HUVECs and fibroblasts showed impaired growth, resulting in gel degradation, BVO's shrinkage, and premature death (Supplementary Fig. 10a). The growth and sprouting of these BVOs, when cultured in wells under identical conditions,

remained unaffected because they were encapsulated in a substantial volume of gel, which protected against any gel degradation in the initial days of culture (Supplementary Fig. 10b).

Although static and flow conditions appeared indistinguishable during the initial days of culture, the flow effects became apparent over time. HUVEC endothelial networks failed to mature properly under static conditions in contrast to flow conditions (Fig. 6a, b, HUVEC-GFP), confirmed by the Angiogenesis Analyzer plugin used for network quantification measurements (Fig. 6c). Additionally, ECM remodeling was substantially enhanced under flow conditions, resulting in the formation of structures that gradually expanded over time in the microchannel adjacent to the trap site. In contrast, ECM remodelling and vascular restructuring was not observed under static culture conditions (Fig. 6a, b, brightfield). Importantly, measurements of the BVOs' diameters between day 0 and day 10 of culture on-chip revealed an enhanced growth of BVOs cultured under flow within an endothelial bed, as compared to the other three conditions (Fig. 6d). In comparing the flow condition to the static condition, it is essential to highlight that the distinctions are not limited to the convective flow and its associated shear; there are also increase in oxygenation and nutrient availability.

We next performed RNA sequencing to assess the gene expression status of 3D tissues cultured without and with flow after extraction from the chip (Supplementary Fig. 11). The four conditions on-chip displayed a notably strong clustering pattern, as evidenced by the dendrogram and principal component analyses (Supplementary Fig. 12). We found 526 genes differentially expressed with a log2[fold change] > 1 and adjusted *P* value below 0.05, comparing the condition static w/o vasc. with the condition flow with vasc. (Fig. 6e), significantly associated with Gene Ontology terms (biological process) related to extracellular matrix and blood vessel development (Fig. 6f). We conducted several control analyses to confirm that these features could be ascribed to our technique, rather than the residual presence of HUVECs (Supplementary Fig. 13). Genes such as *ADAMTS16*, *CAV1*, *ANPEP*, *HMOX1*, *ACE*, known for their key role in angiogenesis, blood vessel development and blood pressure regulation processes[25–31], were found upregulated in condition "flow with vasc." (Fig. 6g). By contrast, genes that are thought to have anti-angiogenic functions or induce cell death in endothelial cells and smooth muscle cells (e.g. *BMP10*, *OLR1*, *AGT*)[32–34], were found downregulated in this condition (Fig. 6g). Moreover, genes of the metalloproteinase family, involved in ECM remodeling[35], were found to be upregulated under flow conditions (Fig. 6g). To explore the impact of static and flow conditions on organoid maturation, we next analyzed the RNA profile of BVOs grown under various conditions on-chip and in conventional 96-well suspension cultures. We then compared these profiles to a reference dataset generated from the Nikolova et al.[36] single-cell RNA-seq data of

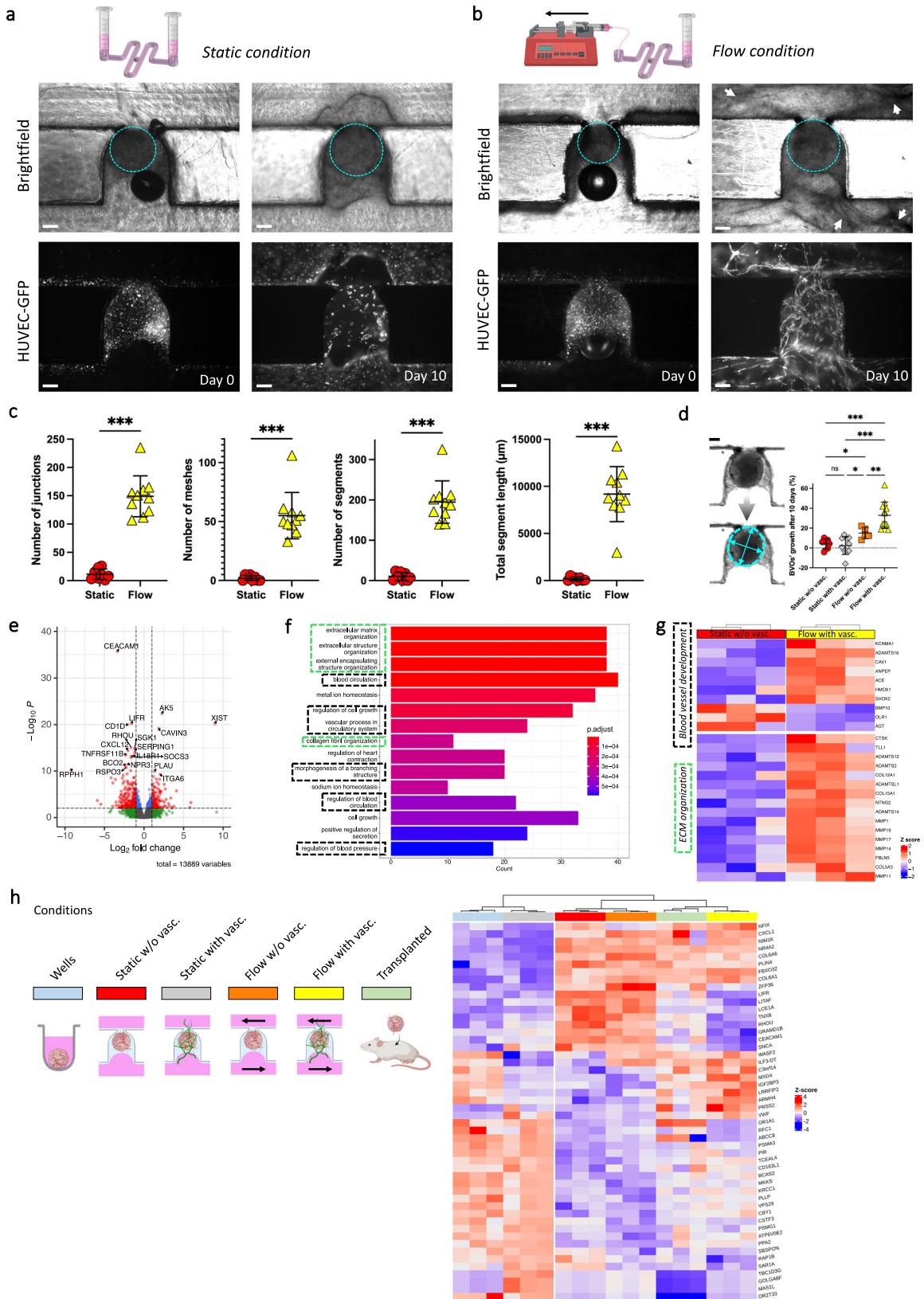

human BVOs matured in mice for two months. Remarkably, we found that the transcriptomes from our samples "flow with vasc." were the most similar to the in vivo transplanted human BVO vasculature (Fig. 6h and Supplementary Fig. 14). These findings indicate that our setup, providing perfusion and flow, promotes blood vessel organoid maturation.

## Enhanced functionality of on-chip vascularized islet spheroids

To further showcase the power of our microfluidic platform, we generated pre-vascularized insulin-secreting pancreatic islet spheroids as a second proof-of-principle of our method. Using the same approach as with the BVOs, we cultured the islets on-chip under static or flow conditions, and with or without a neighbouring GFP-HUVEC

**Fig. 6 | Organoids' growth and maturation enhancement through flow and vascularization. a**, **b** BVO (brightfield, dotted cyan line) and HUVEC network (GFP) development from day 0 to day 10 on-chip, in static (**a**) and flow (**b**) conditions (diagrams created with BioRender.com). Enhanced ECM remodeling under flow conditions can be observed in the brightfield images at day 10, particularly by the presence of black structures surrounding the trap site, indicative of active cell processes (highlighted by white arrows). **c** Angiogenesis Analyzer outputs of the morphology of the endothelial networks between static and flow conditions after 10 days of culture on-chip ($n = 8$ (static) and 10 (flow)). **d** BVOs' growth after 10 days of culture on-chip in various conditions, reported as a percentage increase in their size (average of minor and major axes from the fitted ellipse) compared to day 0 ($n = 7$ (static w/o vasc.), 11 (static with vasc.), 7 (flow w/o vasc.) and 12 (flow with vasc.)). **e** Volcano plot showing differentially expressed genes between conditions static w/o vasc. and flow with vasc. **f** Most enriched (top 15) Gene Ontology (GO) biological process terms resulting from the comparison between conditions static

w/o vasc. and flow with vasc. **g** Heatmap of a selection of genes associated with ECM organization (dotted green boxes) and blood vessel development (dotted black boxes) pathways. The color-coded representation illustrates the expression patterns of key genes involved in these biological processes, enabling a visual comparison of their relative expression levels between conditions static w/o vasc. and flow with vasc. **h** Classification of BVO transcriptomes with a reference tissue simulated from single-cell RNA-seq data of BVOs matured in mice. Semi-supervised classification based on the Kruskal-Wallis test of genes significantly differentially expressed across all conditions, followed by visualization of the top 50 most significant genes (diagrams created with BioRender.com). Scale bars, 200 μm (**a**, **b** and **d**). Data represents mean ± s.d. Statistical significance was attributed to values of $P < 0.05$ as determined by unpaired t test (two-tailed). *$P < 0.05$ ($P = 0.05$ (static w/ vasc. vs flow w/o vasc.) and $P = 0.03$ (static with vasc. vs. flow w/o vasc.)), **$P < 0.01$ ($P = 0.008$), ***$P < 0.001$. Source data are provided as a Source Data file.

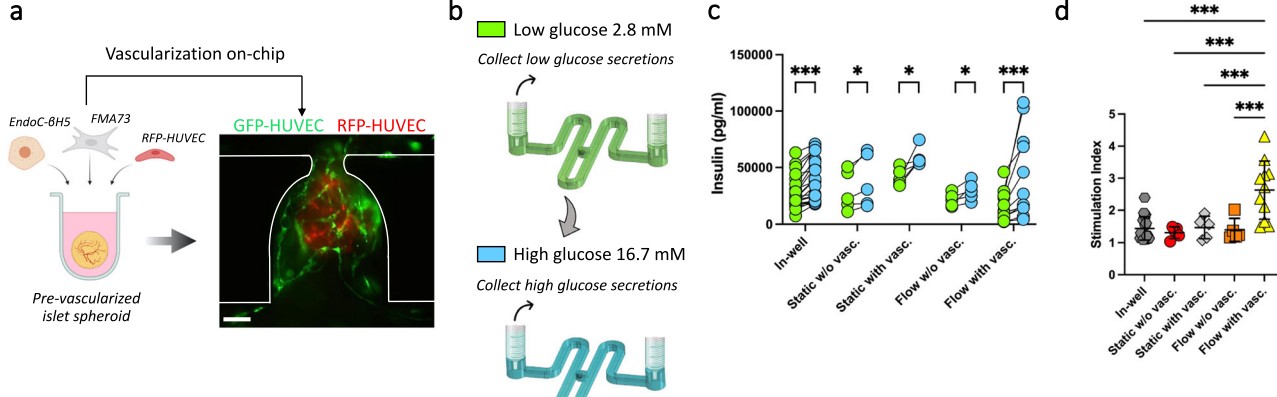

**Fig. 7 | Pancreatic islet spheroids' function enhancement through flow and vascularization. a** Generation of pre-vascularized pancreatic islet spheroids for on-chip culture within a HUVEC endothelial bed and under flow conditions (diagram created with BioRender.com). **b** Schematic diagram of glucose stimulation and collect of secretions performed on-chip (created with BioRender.com). **c**, **d** Comparison of the insulin secretions (**c**) and stimulation index = [high glucose solution]/[low glucose solution] (**d**) of the pancreatic islet spheroids between the various

culture conditions. Each point on the graph represents the stimulation index of one pancreatic islet obtained from the measure of low and high glucose stimulation. Scale bar, 100 μm (**a**). Data represents mean ± s.d. Statistical significance was attributed to values of $P < 0.05$ as determined by unpaired t test (two-tailed). *$P < 0.05$ ($P = 0.017$ (static w/o vasc.), $P = 0.029$ (static with vasc.), $P = 0.037$ (flow without vasc.)), ***$P < 0.001$. Source data are provided as a Source Data file.

endothelial bed (Fig. 7a). At day 5, we performed glucose stimulated insulin secretion (GSIS) assays (Fig. 7b). Insulin concentration measurements at baseline showed no significant differences in all the conditions tested, but insulin secretion was significantly increased in response to high glucose challenge (Fig. 7c). Remarkably, the corresponding stimulation index ([high glucose solution]/[low glucose solution]) was significantly higher for the pancreatic islet spheroids on-chip with flow and with vascularization, compared to all other conditions on-chip, as well as compared to islets cultured in conventional wells (Fig. 7d). The stimulation index for the pancreatic islet spheroids on-chip with both flow and vascularization showed higher variation, which might be attributed to differences in the quality or extent of the vascular network. However, the stimulation index showed no improvement when either only the flow or the HUVEC endothelial bed was introduced. These data indicate an improved insulin secretion in response to glucose stimulation of pancreatic islet spheroids when cultured on-chip under flow conditions and embedded in a HUVEC vascular bed, creating an improved functional β-cell niche.

## Discussion

In this paper we present a microfluidic platform which provides simple, flexible and robust solutions to form vascularized organoids-on-chip with precise control over the fluxes generated during both short- and long-term perfusion. Through this approach, we counteract common barriers in organoid growth, maturation, and organ-on-chip

technologies, such as manual operation and a lack of automation in construct location and hydrogel handling. First, the device was manufactured using COC, a material readily adapted by industry, avoiding fundamental complications of PDMS systems such as incompatibility with hydrophobic compounds or small molecule absorption. Second, in most studies, the flow rate is determined by hydrostatic pressure, a simple and cost-efficient liquid actuation principle. However, optimal hydrostatic differential pressure cannot be maintained for a long time, with the consequence that the culture medium must be changed regularly. This can be avoided by using rocker perfusion platforms[37,38], at the consequence of a reduction in physiologically-relevant flow mechanics (i.e. pressure modulation and/or peristalsis). In our study, this has been overcome through the use of a syringe pump. Using 20 ml syringes, the microfluidic perfusion can be maintained for two weeks at a flow rate of 1 μl/min without disconnecting the syringes. It should be noted that by using a 10-channel syringe pump, as was done in this study, flow rates in 10 different channels can be readily monitored in parallel. Third, the organoids and hydrogels were passively placed into a hydrodynamic trap, wherein the cells were subjected to minimal shear stress. These traps can be put in series, thus hosting multiple vascularized organoids (including different tissue organoids) interconnected through endothelial networks.

Cell and hydrogel loading, as well as long-term organoid perfusion with a defined growth medium, were performed through a single fluidic inlet. Our automation of fluid handling in this system facilitates

widespread use, distribution, and adaptation to microfluidic devices in both industries and healthcare settings. The presented microfluidic platform offers robustness through an innovative encapsulation technique, which relies on hydrodynamic and capillary effects that are largely independent of the working pressure. While gel loading is often a delicate step in conventional chip designs, the process used in this study is highly reproducible and easy to perform. The loading steps can be performed in seconds, allowing all operations to be completed at room temperature. With flexibility in application to a variety of bioengineering projects, this microfluidic platform supports the vascularization of nearly any 3D construct simply by adjusting perfusion channel and trap dimensions. In summary, our organ-on-chip platform is robust, flexible, and user-friendly, overcoming key hurdles in the industrialization of organ-on-chip models and providing access to user-friendly organoid-on-chip technologies.

Several limitations of our platform merit discussion. First, the encapsulation process of endothelial cells and fibroblasts in the hydrogel results in a significant cell loss when the excess gel is expelled before polymerization. This becomes particularly pronounced for larger designs intended for large organoids, where the gel volume, and consequently the number of cells required, escalates considerably. This poses challenges, especially when considering more valuable cell types like isogenic iPSC-derived vascular cells. Furthermore, while adjusting the gel layer thickness around the organoid is theoretically feasible, its practical execution is intricate. We invested significant effort into modeling the Landau-Levich-Bretherton phenomenon, which dictates the residual thin gel layer on the microchannels post air bubble passage. Our setup's unique design adds to this complexity, and addressing these challenges remains a priority as we refine the platform.

Nevertheless, our microfluidic device supports the vascularization and perfusion of 3D biological tissues. The formation of perfusable endothelial networks was found to be highly reproducible. We were able to generate physiological fluxes in these networks, with observed velocities ranging from $v = 100\,\mu m/s$ to $v = 7500\,\mu m/s$ and shear rates ranging from $\dot{\gamma} = 25\,s^{-1}$ to $2000\,s^{-1}$. Thus, physiological blood flow velocities and shear stress dynamics observed in the human body can be readily achieved using our microfluidic platform—an advantage over commonly reported hydrostatic pressure operated devices, that result in lower flow rates[39]. Finally, the deposition of the hydrogel on the walls of the microchannels, conditioned by the Landau-Levich-Bretherton effect, offers a robust means of achieving continuous endothelial cell lining of the perfusion channel, recapitulating aspects of vascular patterning of the tunica intima in vivo.

The chip configuration described here resulted in anastomosis between a HUVEC endothelial bed and the capillaries of human blood vessel organoids. Consequently, our system can generate perfused hierarchical networks that encompass vessels ranging in size from arterioles to venules and capillaries. Most importantly, we have now accomplished intravascular perfusion of blood vessel organoids, thus making our microfluidic platform the first device to incorporate functional vasculature throughout microfluidic-trapped embedded 3D organoid constructs. Our platform now allows us to explore diverse topics, such as organoid lifespan enhancement through vascularization, exposure to drugs, nucleic acids or metabolic stress. The device we have developed also offers the flexibility to vascularize other types of organoids, spheroids, tumoroids, or human tissue explants, as exclaimed in our study by improved glucose responsiveness of islet spheroids.

## Methods
### Microfluidic device fabrication
Computer-aided design (CAD) files of the chip were created using SolidWorks 2022 (Dassault Systèmes). The microfluidic chips were made of Cyclic Olefin Copolymer (COC) because of its low autofluorescence, strong chemical resistance, and low drug absorption. Microfluidic patterns were directly machined on a COC sheet (TOPAS, USA), using high-precision milling (DATRON M7HP equipment). The chip (84 mm×54 mm) contained 10 identical microfluidic circuits. The square-profiled channels were 400 μm x 400 μm for the experiments with mesenchymal spheroids, and 800 μm x 800 μm for experiments with blood vessel organoids. The microfluidic channels were sealed with a MicroAmp optical adhesive film (Applied Biosystems, cat. no. 4306311).

### Trapping mechanism and cell seeding procedure
The microfluidic setup is initiated using a syringe at the channel entrance, to sequentially introduce: (1) non-polymerized hydrogel embedding spheroids/organoids and endothelial cells, (2) air, and (3) growth medium. The channel features a serpentine loop that is bypassed by a U-cup shaped microchannel, which serves as a cell aggregate trap. When unoccupied, the hydraulic resistance of this trap ($R_1$, Path A) is less than that of the serpentine loop ($R_2$, Path B), thereby guiding spheroids/organoids into the trap by flow preference.

To initiate the seeding, a spheroid/organoid was gently extracted from a 96-well plate via a pipette tip and mixed into the hydrogel with thrombin. This mixture, amounting to 50 μl and containing the spheroid/organoid, was placed into the reservoir leading to the microfluidic system. Using a syringe pump set to a withdrawal mode with a flow rate of $Q = 300\,\mu l/min$, the spheroid/organoid progressed through the channel and was captured by the U-cup microchamber. Subsequent introduction of air served to position the hydrogel before it solidified. The Hydrogel remained at the U-cup, securing the spheroid/organoid and lining the channel corners. Both the hydrogel mix (prior to thrombin addition) and thrombin were kept at 4 °C until used for the experiment. Immediately after mixing thrombin with the fibrin hydrogel, it was loaded onto the chip. This microchannel injection process takes approximately 10 seconds, largely excluding viscosity changes in the gel.

### Cell culture and generation of spheroids and organoids
**Mesenchymal spheroids.** Primary human fibroblasts (FMA73) were extracted from skin explants obtained through the elective breast surgery of a healthy young woman following informed consent; this tissue was provided by Walid Rachidi, CEA Grenoble. GFP- and RFP-labelled HUVEC cells (Angio-Proteomie, cat. no. CAP0001GFP and cat. no. CAP0001RFP, respectively) were cultured in complete EndoGM medium (Angio-Proteomie, cat. no. CAP02). Passage 5−7 cells were used for the experiments. Fibroblasts cultured in Fibroblast Growth Medium-2 (Lonza, cat. no. CC-3132), and passage 6−8 cells were used for the experiments. We prepared fibroblasts and HUVEC co-culture, termed mesenchymal spheroid model here, in U-shaped 96-well ultra-low attachment microplates (Corning, cat. no. CLS4515). Fibroblasts and HUVEC cells were mixed at a ratio of 1:1 (5000 cells per well) in 150 μl of medium consisting of a mix of CnT-ENDO (Cellntec, cat. no. CnT-ENDO) / CnT-Prime Fibroblast medium (Cellntec, cat. no. CnT-PR-F) at a ratio 1:1. After pre-culturing for 1 day in the microplate, a spheroid was introduced into the device. The same medium mix of CnT-ENDO / CnT-Prime Fibroblast medium was used for the microfluidic perfusion of the fibroblasts and HUVEC co-culture spheroids. RFP-HUVEC cells were suspended in the hydrogel at a concentration of $6\times10^6$ cells per ml.

**Blood vessel organoids (BVOs).** 3D human blood vessel organoids were generated from human induced pluripotent stem cells (hiPSCs) as previously described[17]. In brief, NC8 stem cell colonies were harvested using Accutase (Gibco, cat. no. A1110501) to get a single cell suspension. To make sure that the organoids would be of an appropriate size to fit into the 800 μm x 800 μm square profile microchannels of the chip used for BVO experiments, AggreWell™400

(STEMCELL Technologies, cat. no. 34415) plates were used. Each well of the plate contains 1200 microwells with a 400 μm diameter. 600000 single stem cells were seeded per well (500 cells/microwell) in aggregation media with 50 μM Y-27632 (Tocris, cat. no. 1254/10). Mesoderm induction and sprouting was induced directly in the AggreWell™400 plates by carefully changing the media with a p1000 pipette making sure to not disturb the cell aggregates in the microwells. For Collagen I-Matrigel embedding, organoids were harvested from the AggreWell™400 plate by vigorously pipetting up and down with a cut p1000 tip close to the bottom of the well. Harvested organoids were embedded in a 12-well plate (approx. 100 organoids/12-well), and subsequently cut out and singled into low attachment 96 U-well plates 4-5 days after embedding as previously described[17]. No alterations were made to any of the BVO differentiation media. The BVOs were maintained in a differentiation medium containing 15% FBS (Sigma-Aldrich, cat. no. F1051), 100 ng/ml VEGF-A (PrepoTech, cat. no. 100-20) and 100 ng/ml FGF-2 (Miltenyi Biotec, cat. no. 130-093-564). Organoids with a diameter of 500-600 μm were selected and added to the microfluidic chip at day 15. A mix of the differentiation medium with CnT-ENDO and CnT Prime Fibroblast media was used for the long-term culture on-chip (ratio 1:1:1). To address concerns regarding the stability of sensitive growth factors in our cell culture media, such as FGF-2 and VEGF-A, we replenished the syringe reservoirs with fresh media daily.

**Pre-vascularized pancreatic islet spheroids.** Commercially available human primary β cells called EndoC-βH5 (Human Cell Design) were co-cultured immediately after thawing with the above-mentioned primary fibroblasts FMA73 and HUVEC-RFP in a U-shaped 96-well microplate with an ultra-low attachment surface, in a mix of CnT-ENDO, EndoC-βH5 culture medium (Human Cell Design, cat. no. ULTIB1-100) and DMEM/F-12 (Gibco, cat. no. 11330032) with 1% Penicillin-Streptomycin (Gibco, cat. no. 15140122). A ratio of 6:1:1 (EndoC-βH5:FMA73:HUVEC-RFP, 10 000 cells per well) was used, resulting in 300 μm spheroids after 6 days. Passage 5-12 cells were used for these experiments.

### Hydrogel preparation
A fibrin-hydrogel made of 6.6 mg/ml fibrinogen (Sigma-Aldrich, cat. no. 9001-32-5), 0.15 TIU/ml aprotinin (Sigma-Aldrich, cat. no. 9087-70-1), 2.5 mM CaCl$_2$ (Sigma-Aldrich, cat. no. 10035-04-8), and 1 U/ml thrombin (Sigma-Aldrich, cat. no. 9002-04-4) prepared in HEPES-buffered saline was used in all experiments. After adding the thrombin into the mixture, all the procedures were quickly performed to avoid premature gelation.

### On-chip immunofluorescent staining
For immunofluorescent staining, the tissues were fixed by flowing 4% paraformaldehyde (Boston BioProducts, cat. no. BM-155) for 1 h at room temperature through the microchannel, and subsequently blocked with 3% FBS (Sigma-Aldrich, cat. no. F1051), 1% BSA (Sigma-Aldrich, cat. no. A9647), 0.5% Triton-X-100 (Sigma-Aldrich, cat. no. T8787) and 0.5% Tween (Sigma-Aldrich, cat. no. P7949) for 2 h at room temperature. Primary antibodies CD31 (Abcam, ab134168, rabbit anti-human, 1:200), CD31 (Abcam, ab9498, mouse anti-human, 1:200), PDGFRβ (Cell Signaling Technology, 3169 S, rabbit anti-human, 1:200), ColIV (Chemicon, AB769, goat anti-human, 1:50), SM22/TAGLN (Abcam, ab14106, rabbit anti-human, 1:200), ZO-1 (Abcam, ab216880, rabbit anti-human, 1:200), VE-Cadherin (Abcam, ab33168, rabbit anti-human, 1:200), were diluted in blocking buffer and flowed overnight into the microfluidic chip at 4 °C. After a 30 min wash in PBST (0.05% Tween), secondary antibodies (donkey anti-mouse Alexa Fluor 488, Invitrogen, A-21202, donkey anti-rabbit Cy3 Jackson ImmunoResearch Inc., 711-165-152, donkey anti-rabbit Alexa Fluor 555, Invitrogen, A-31572, donkey anti-rabbit Alexa Fluor 647, Invitrogen, A-31573, donkey anti-goat Alexa Fluor 555, Invitrogen, A-21432, donkey anti-goat and

Alexa Fluor 647, Invitrogen, A-21447) were flowed into the microchannels at 1:200 in blocking buffer for 2 h at room temperature. After a 30 min wash in PBST, nuclear counter-staining using Hoechst 33342 was carried out according to a routine protocol.

### Imaging
The microfluidic chip was imaged on a daily basis using an inverted Olympus IX50 microscope for the period of the experiment. Images were taken in brightfield and fluorescence channels, with 5x and 10x objectives. After immunostaining, the microchannels were imaged using either a Nikon Eclipse Ti-E Spinning Disk microscope, a Zeiss LSM880 scanning confocal microscope or a Leica SP8 scanning confocal microscope, with 10x, 20x, 25x, 40x and 63x objectives. The flow in vascular networks was assessed in the second week of culture by loading polystyrene fluorescent microbeads (Thermo Fisher Scientific, Fluoro-Max Fluorescent Beads) into the serpentine channel. Images were captured at 15 Hz using the inverted Olympus IX50 microscope described above. Microbeads were tracked in perfused tissues from separate microfluidic channels using Fiji (ImageJ). For microbeads perfusion in the BVOs' vasculature, since we needed to capture images deep within the tissue at a high frame rate while minimizing photobleaching, confocal microscopy was unsuitable. Therefore, we employed a Leica THUNDER 3D Cell Imager microscope for this purpose.

### Light sheet fluorescence microscopy
A homemade light sheet fluorescence microscope was used in this project, which we adapted to image biological samples inside microfluidic chambers without interfering with the normal function of the chip[40]. The light sheet was generated with a 488 nm Ar-laser, focused by a 100 mm focal length cylindrical lens. The fluorescence signal generated at the illuminated plane was collected by a long working distance, with the objective (Mitutoyo M Plan APO SL 20X, 0.28 N.A.) placed at 90° to the excitation path. The sample plane was at 45° from both paths. A tube lens was associated to the objective to form the image of the fluorescent structure onto a high-sensitive sCMOS camera (Hamamatsu HPF6 ORCA FLASH 4.0 V3) with a magnification factor of 12. To filter out the laser excitation, a high pass (cut-off wavelength of 490 nm) interference filter was used. The sample was mounted onto a custom-designed holder attached to computer-controlled xz linear translational stages. In this configuration, the microfluidic chip was kept horizontal, and the thinner lateral part of the light sheet was positioned at the surface of the gel. The light sheet illuminated the sample in the direction perpendicular to that of the microfluidic channel (Supplementary Fig. 3b-d).

### Endothelial networks analysis
Confocal z-stacks of the microchannels in various culture conditions were taken. These stacks were then flattened in ImageJ to a 2D maximum intensity projection and analysed using the Angiogenesis Analyzer plugin with default settings[41]. Four metric parameters were selected for this study, namely the number of junctions, the number of meshes, the number of segments and the total segments length.

### Imaris analysis
Confocal z-stacks were opened and 3D-rendered using Imaris imaging software. HUVEC endothelial networks and BVOs' vasculature were rendered as surfaces after masking with the GFP-HUVEC and CD31 signals, and the fluorescent microbeads as spots of known diameter.

### RNA sequencing
We performed bulk RNA sequencing to investigate gene expression profiles of the samples under scrutiny. Using a needle, BVOs were meticulously extracted from the chip, with special care taken to isolate

the BVO without including the surrounding proliferating tissue. RNA extraction was done using the Trizol protocol (Invitrogen). Sample quality control was performed using the Agilent 2100 Bioanalyzer to ensure high-quality RNA input. Qualifying samples underwent library preparation following the standard protocol for the Illumina Stranded mRNA prep (Illumina). Subsequently, sequencing was performed on the Illumina NextSeq2000 platform, generating 59 bp × 59 bp Paired End reads. Post-sequencing, the data was demultiplexed using Illumina's BCL Convert, and the resulting de-multiplexed read sequences were aligned to the Homo sapiens (hg38 no Alts, with decoys) /Mus Musculous (mm10) reference sequences. Alignment was carried out using the DRAGEN RNA app on the Basespace Sequence Hub.

### Bioinformatics analysis

**Pre-processing of bulk transcriptome data.** Gene expression profiles measured as raw read counts were first filtered for low expressed genes keeping only genes whose sum of reads across all samples was greater than or equal to 10. Then the ENSEMBL gene ID were converted into gene symbols using the mapIds() function of the R/Bioconductor package AnnotationDbi (v.1.60.0). Only genes with a gene symbol and with a total sum of reads across all conditions greater than or equal to 10 were kept for further analysis. Gene expression values were then normalized with the VST method from the R/Bioconductor package DESeq2 (v.1.3.8.3).

**Differential gene expression analysis.** The identification of differentially expressed genes (p-value < 0.05) across multiple groups of experimental conditions was performed with the non-parametric Kruskal-Wallis rank test from normalized gene expression values using the col_kruskalwallis function of the R/Bioconductor package matrixTests (v.0.1.9.1). The list of differentially expressed genes was visualized using the R/Bioconductor package ComplexHeatmap (v.2.14.0) and these parameters: z-score normalization by row and by column, use of Pearson as distance correlation method and use of ward.D2 as clustering method. Sample pairwise identification of differentially expressed genes was performed with R/Bioconductor package DESeq2 (v.1.3.8.3) (FDR corrected p-value < 0.05; absolute Log2 fold change > 1; baseMean > 18).

**Pathway enrichment analysis.** The ontological enrichments were performed from the lists of differentially expressed genes with the R/Bioconductor package clusterProfiler (v.4.6.0) (p-value < 0.05; Benjamini-Hochberg q-value < 0.2; minGSSize = 30; maxGSSize = 500). The reference gene vector used as background corresponded to the genes expressed in the samples considered for the analyses.

**Generation of pseudobulk mixture for reference tissue.** From a gene read count matrix of single-cell RNA-seq data of 16,410 cells of BVOs transplanted into mice for their maturation, we randomly selected three times 5000 cells without replacement. For each batch of 5000 cells, the sum of reads per gene across cells was performed to generate a pseudobulk sample. These 3 pseudobulk samples were then integrated with the other experimental conditions and filtered in the same way. To perform comparisons between transcriptomic data of different types, bulk RNA-seq and pseudobulk mixture generated from single-cell RNA-seq data, we removed technical biases and adjusted artefactual variations in gene expression using the RUVg function from the R/Bioconductor package RUVSeq (v.1.32.0). The list of negative control genes, having constant expression across all experimental conditions, was obtained empirically using the non-parametric Kruskal-Wallis rank test applied to all conditions. Non-differentially expressed genes, associated with a p-value > 0.10, were used as negative control genes. Concerning the number of undesirable factors k, the higher the value of k, the greater the probability of deleting the biological signal of interest. We therefore selected the value k = 1

because beyond this threshold we could begin to detect the non-grouping by clustering of some biological replicates.

### Glucose stimulated insulin secretion (GSIS) assays

Pre-vascularized pancreatic islet spheroids were individually placed in a 96 well plate with ultra-low attachment surface, and incubated at 37 °C, 95% O2, 5% CO2 for 1 h in 60 μl KREBS buffer (Sigma-Aldrich, cat. no. K4002), 1% BSA, 2.8 mM glucose (Gibco, cat. no. A2494001) as pre-incubation. Thereafter, islets were incubated for 1 h with 2.8 mM glucose solution (low glucose solution) and for 1 h with 16.7 mM glucose solution (high glucose solution). After each incubation step, supernatants were collected and stored at −80 °C. Insulin concentration in each collected supernatant was measured using STELLUX Chemiluminescence Human Insulin ELISA (ALPCO, cat. no. 80-INSHU-CH01). Similar protocol was performed for on-chip experiments, following a protocol previously described[42]. Each sample was measured in duplicate. A stimulation index was obtained by calculating the ratio of insulin measured between high and low glucose stimulation, [high glucose solution]/[low glucose solution].

### Statistics and Reproducibility

Results are shown as mean ± s.d. as indicated in the Figure legends. Statistical analyses were conducted using GraphPad Prism 9 (GraphPad Software Inc.). Different significance levels (P values) are indicated in each figure with asterisks (*P < 0.05, **P < 0.01, ***P < 0.001) and exact P values when possible.

For transparency, we state here the number of biological samples on which experiments were repeated independently, with similar results obtained, to produce the data shown: Figs. 1h, 6; Fig. 2a, b, > 50; Fig. 2c, 12; Figs. 3b, 6; Fig. 3c, 10; Fig. 3d, > 50; Fig. 4b, 30; Figs. 5b, 6; Figs. 5e–g, 6; Figs. 5i, j, 6; Fig. 6a, b, at least 8 for each condition; Figs. 6e–h, 3 replicates for each condition; Figs. 7c, d, 22, 5, 5, 5, and 11 replicates per condition (wells, static w/o vasc., static with vasc., flow w/o vasc., and flow with vasc. respectively).

### Reporting summary

Further information on research design is available in the Nature Portfolio Reporting Summary linked to this article.

## Data availability

Bulk RNA-seq data have been deposited with the accession number "PRJNA1061525". All other data supporting the findings of this study are available within the article and its supplementary files. Any additional requests for information can be directed to, and will be fulfilled by, the corresponding authors. Source data are provided with this paper.

## Code availability

All the codes used for this study are available at: https://github.com/ClementQuintard/A-microfluidic-platform-integrating-functional-vascularized-organoids-on-chip.

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

## Acknowledgements

We thank N. Verplanck and F. Boizot for the manufacturing of microfluidic chips, and X. Mermet for his help with light sheet microscopy experiments. We thank M. Nikolova and B. Treutlein for sharing with us their sc-RNA sequencing data. This work was supported by the CEA "OOC inflexion" and received funding from GRAL, a program from the Chemistry Biology Health (CBH) Graduate School of University Grenoble Alpes (ANR-17-EURE-0003). Imaging was performed in the LSI Imaging Core Facility of the Life Sciences Institute at the University of British Columbia, supported by Life Sciences Institute, the UBC GREx Biological Resilience Initiative. The infrastructure within LSI Imaging Core Facility is funded by the Canadian Foundation of Innovation, BC Knowledge Development Fund, Natural Sciences and Engineering Research Council Research Tools and Instruments, and UBC Research Facility Support Grants as well as a Strategic Investment Fund (Faculty of Medicine, UBC). C.Q. and J.M.P. were funded by the Leducq Foundation with the Transatlantic Network of Excellence grant "ReVAMP — Recalibrating Mechanotransduction in Vascular Malformations" (2022–2027). G.J. and J.M.P. received funding from the Vienna Science and Technology Fund (WWTF) [10.47379/EICOV20002] and the Fundacio La Marato de TV3 (202125-31). Research in the lab of J.M.P was further funded by the Medical University of Vienna, the Austrian Federal Ministry of Education,

Science and Research, the Austrian Academy of Sciences, the T. von Zastrow foundation, the Canada 150 Research Chairs Program F18-01336, the Canadian Institutes of Health Research COVID-19 grants F20-02343 and F20-02015, an Allen Distinguished Investigators (ADIs) award (AWD-020087 PGAFG 2021), the German Federal Ministry of Education and Research (BMBF) under the project "Microbial Stargazing—Erforschung von Resilienzmechanismen von Mikroben und Menschen" (Ref. 01KX2324), and the Innovative Medicines Initiative 2 Joint Undertaking (JU) under grant agreement No 101005026. This Joint Undertaking receives support from the European Union's Horizon 2020 research and innovation program and EFPIA. D.A.L and J.M.P acknowledge support from the UK-Canada (MRC-SCN) regenerative medicine exchange programme and a UK-Canada Diabetes Research Team Grant (MR/T032251/1).

## Author contributions

C.Q. performed the on-chip experiments using mesenchymal spheroids and blood vessel organoids. E.T. performed the on-chip experiments using the pancreatic islets spheroids. C.Q., G.J., J.W., N.W. and A.L. generated the blood vessel organoids. C.Q. and C.L. generated the mesenchymal spheroids. Emily Tubbs and Amandine Pitaval generated the pancreatic islet spheroids. C.Q. and J.J. designed the RNA sequencing experiments and performed the RNA extraction. C.Q., T.-S.B., G.P. and C.B. performed the bioinformatics analysis. C.B. and P.B. designed and built the light sheet microscope, and C.Q. and C.B. performed the experiments using this set-up. C.Q., J.-L.A. and Y.F. designed the microfluidic chips. J.K. manufactured the microfluidic chips. A.H., D.A.L., F.N., Y.F., J.M.P. and X.G. conceived and supervised the project. C.Q., Y.F., J.M.P. and X.G. wrote the manuscript with inputs and comments from all coauthors. All authors contributed to data interpretation and finalization.

## Competing interests

C.Q., J.-L.A. and Y.F. are the authors of the patent application US20210277349A1 (granted) describing the method for microfluidic perfusion used in this study. J.M.P. declares a conflict of interest as he is one of the authors of the patent application US20200199541A1 relating to blood vessel organoids generation. Furthermore, J.M.P. is the founder, shareholder, and chairman of the scientific advisory board of Angios Biotech, a company establishing BVOs for drug testing and vascular transplants. The remaining authors declare no competing interests.
