## [Peer Review File · Nature Communications]

REVIEWER COMMENTS

Reviewer #1 (Remarks to the Author):

In this paper, Quintard et al develop a microfluidic approach to vascularize organoids, and show its application in generating and perfusing HUVEC-based endothelial networks around mesenchymal and pancreatic islet spheroids, as well as in linking this HUVEC-based network with a vascular organoid. They show that these experiments remain viable up to 15 days in culture, and that some organoid characteristics are enhanced as a result of this perfusion. There have been multiple perfusion and vascularization platforms reported in the last few years, and this one interesting and novel in that it (1) repurposes a rather classic system of bead/single cell microfluidic trapping to trap larger sized organoids in a set of microfluidic traps and (2) combines this with the deposition of a endothelial cell-laden hydrogel, to efficiently (and presumably consistently, though this should be shown) coat the walls of the chip and crucially, a part of the organoid. Conceptually, this represents an elegant and new technical approach to the organoid vascularization problem, which would allow for robustness and scalability. As such, this platform represents a really promising technical advance for the field, and shows some interesting biological applications, particularly that gene expression of blood vessel organoids (BVOs) in flow and vascularized conditions on chip is closest to that of organoids transplanted in vivo, and that there are functional consequences to flow to flow with vascularization for glucose stimulated insulin secretion.

There are nonetheless four important issues with this paper, which would have to be addressed:

(1) Some of the novelty that the authors claim regarding their platform is overemphasized – indeed a number of systems exist which already vascularize organoids (including with some level of perfusion) – so the authors would have to be more clear on this point.

“However, this conventional geometry does not allow reproduction of the fluxes observed in vivo, and no success on truly vascularized organoids has been reported yet.”

What do the authors mean by truly vascularized? Penetration of flow the inner core of the organoid has already been shown in multiple systems.

“Moreover, to date, no in vitro system has been convincingly shown to establish intravascular perfusion within organoids through functional anastomosis with neighbouring endothelial networks.”

Again this has been done, for example:

Integrating perfusable vascular networks with a three-dimensional tissue in a microfluidic device

Nashimoto et al, 2017

“With flexibility in application to a variety of bioengineering projects, this microfluidic platform supports the vascularization of nearly any 3D construct simply by adjusting perfusion channel and trap dimensions.”

This statement should be toned down. Only three systems were tested, and success rates can differ between organoid systems.

(2) One of the key advantages I see in this platform is the robustness, scalability and versatility of the platform – although other platforms may be able to achieve similar vascularization, this seems like it would be much faster and reliable - the authors should emphasize this point by showing and quantifying the variability of trapping efficiency, vascularization, both within a single chip as well as between chips.

The text related to the organoid trapping methodology could be more clearly explained.

How is the trapping of the organoid occurring in fibrin which is presumably changing viscosity, and actually solidifying? Is the chip kept on ice or is there a limit of time in which this process can occur ?

The efficiency of trapping should be quantified. How many organoids are typically trapped in a chip and does this vary between chips. One of the main advantages of this approach, as I see it, is the apparent robustness of the process, but this should be quantified and shown.

Additionally, it is unclear how many organoids were assessed in Fig. 3.

(3) Fibrin is used as the supportive hydrogel in all experiments, however fibrin used far less in organoids than Matrigel or synthetic gels. Is the trapping methodology also compatible with other gels? It would be important for the authors to show that this approach is more broadly applicable than just with fibrin, by that this approach is compatible at least with Matrigel, the standard gel used with organoids, and preferably with an additional gel system.

Additionally, can the gel layer thickness around the organoid be varied ? I.e. is the vascular network limited to the thin layer around the organoid?

(4) Although the biological results are compelling, in some cases (BVOs) key controls are missing

For example,

“BVOs cultured on-chip in the fibrin hydrogel without HUVECs and fibroblasts show impaired growth, resulting in gel degradation, BVO’s shrinkage, and premature death (Supplementary Fig. 10).”

Important control is missing: BVOs in fibrin in wells i.e. off-chip – to understand if it’s the fibrin that is inhibiting BVO growth.

“Fibroblasts, in comparison to HUVECs and hPSC-derived cell populations, produce more extracellular matrix (ECM), thereby preserving the gel structure. In the absence of fibroblasts, ECM production is

limited, resulting in reduction in overall matrix and scaffolding support for the growth, development, and maturation of new vessel networks.”

This doesn't really make sense when one compares these results to very effective BVO growth off-chip in suspension without fibroblasts, but importantly without fibrin matrix. Additionally, BVOs include pericytes – can these be evidenced in these experiments, and do they have a role in maintaining the integrity of the vascular network?

Additional comments:

-The ability to address each microchannel independently is a distinct advantage of the platform, which is clear in the text and supplementary figures but not so much from Figure 1. This could be emphasized graphically.

-It is mentioned that organoid size is 300-600um – the authors should show the range of organoid sizes which can be trapped in the current platform.

-Experiments are only carried out to D13-D14. What happens thereafter? The writing is ambiguous here: “network-like structures with a three-dimensional configuration developed and remained stable up to day 13 (Fig. 2d).”

-“fluid velocities ranging from $v_{min} = 100 \mu\text{m/s}$ to $v_{max} = 7500 \mu\text{m/s}$ (Fig. 2i). Assuming laminar flow,”

An approximation of the Reynolds number should be calculated and ascertained to be laminar. These channels seem to be larger than in conventional microfluidics, so it is quite possible that turbulent flow may be occurring, which may explain potentially explain some recirculating regions seen in the bead experiments.

-Why do the beads have preferential paths even outside the organoid, i.e. in the channels? There seem to be some high concentrations at some locations and it's not clear if these are zones of recirculation, blockage or something else. Can the authors explain these phenomena?

-Another missing (on-chip) control is : only BVO seeded without HUVEC cells to check if the microbeads cannot penetrate the BVO itself, i.e. whether the link with the HUVEC network is necessary for perfusion to occur.

-What genes were still differentially expressed between in vivo transplantation and in vitro on chip vascularization? It would be interesting to discuss this since it may suggest what is still needed to improve the system.

-The results of the glucose stimulated insulin secretion are compelling, however there is high variability in glucose response in the perfused system. Can the authors make a retrospective analysis of the vascular networks to determine if conditions where a “better” vascular network is associated to the enhanced insulin secretion?

Reviewer #2 (Remarks to the Author):

In this manuscript the authors present a microfluidic platform incorporating perfused vasculature and organoids. While the paper is well-written, much of the work is carefully performed, and the authors present some interesting methods, overall the work lacks significant novelty and the conclusions are somewhat overstated and not fully supported by the data.

The authors claim “Importantly, we demonstrate effective anastomosis and controlled perfusion of the vascular organoids, as well as enhanced organoid growth, maturation, and vasculature development”. Firstly, these “organoids” would better be described as spheroids as, with the exception of figure 6, they are not modeling any actual organs. Secondly, “growth and maturation” of organoids representing a specific tissue would be of considerable interest, however in this case the growth and maturation appears to refer to the formation of vascular networks – not a novel finding. Thirdly, a significant portion of the data show perfusion of beads through the organoids, demonstrating that HUVECs and fibroblasts cultured in a fibrin gel under flow generate perfusable networks. This has been demonstrated many times by several different groups and again, is not at all novel. Other laboratories that have published substantially similar findings include those of Kamm, Chen, Hughes, Jeon, George, Huh, among others. In addition, there is a world of difference between perfusable vessels and tissue perfused exclusively through the vessels. The latter, obviously, represents the in vivo state but ,many of these models have significant leak around (not out of) the vessels, which makes interpretation of their significance to the tissue very hard to determine. So, in this study, is the flow exclusively through the vessels? Do the vessels leak and is this leak at physiological levels? Also, with regard to the vessels, what diameter do they have and do they form hierarchical networks? Are they wrapped by pericytes, smooth muscle cells or both? These are critical components of organ vasculature and need to be determined for these “organoids”.

The authors generated three organoid models: HUVECs and fibroblasts cultured for several days in a fibrin hydrogel; iPSC derived ECs cultured with fibroblasts in a fibrin hydrogel; and, a beta cell line cultured with fibroblasts and HUVECs in a fibrin hydrogel. All organoids were coated further in Fibrin+HUVECs+fibroblasts prior to being loaded into a custom microfluidic platform with a trap to catch the organoid. Following are some more specific concerns regarding the experiments presented:

Figure 1-3: the HUVECs+fibroblast model. The authors demonstrated that under flow, vasculature in their organoid connects with surrounding vessels. As noted above, dye perfusion of different molecular weights would be very helpful as an indicator of the “quality” of the vasculature and whether it is leaky. The authors also state that “The chip configuration described here resulted in anastomosis between a HUVEC endothelial bed and the capillaries of human blood vessel organoids. Consequently, our system can generate perfused hierarchical networks that encompass vessels ranging in size from arterioles to venules and capillaries.” Beside CD31 staining and network length, the authors don't investigate the hierarchical structure of their networks, nor do they examine endothelial artery/vein markers, nor do they demonstrate pericyte or SMC coverage, so it is hard to see how they can claim to have a hierarchical network comprised of arterioles/venules/capillaries.

Figure 3-4: iPSC-EC derived vascularized organoids encapsulated with HUVECs. The authors demonstrated that when co-cultured with HUVECs under flow, their iPSC-EC derived vessels will lumenize and anastomose with HUVEC-derived vessels. An important control here would be BVO encapsulated with fibrin containing iPSC-ECs not HUVECs. It is difficult to determine if the iPSC-ECs are maturing due to signaling from HUVECs or if the chip is playing a key role in iPSC-EC vessel maturation. In addition, although the authors say “Numerous BVOs vessels (CD31+GFP-) were observed near HUVEC networks, indicating functional perfusion from the HUVEC endothelial bed into the organoid vessels (Fig. 4b and Supplementary Fig. 7)”, there do not appear to be any data actually showing perfusion (beads, for example).

Figure 5: This figure compares their BVO's covered in fibrin either with or w/o ECs & fibroblasts. The authors demonstrated that without including HUVECs and fibroblasts in the fibrin coating, the BVO's begin to die and shrink. This is unsurprising given that the role of fibroblasts in supporting vasculature in fibrin gels is well documented. The scRNAseq analysis seems to be very preliminary. The finding of matrix organization under flow is fully expected and not of particular interest. The endothelial cells and stroma should be analyzed independently and cross-talk between the populations could be examined. Are subpopulations of EC seen – for example, artery, vein and capillary? The comparison between the organoids in vitro and in vivo is an understandable choice, but of far more interest would be a comparison to endogenous vasculature/stroma in vivo.

Figure 6: The authors set out to demonstrate the actual utility of the device for supporting a pancreatic islet model, however this feels very under-developed. The authors generate their “islets” by culturing a commercially-available beta cell line with HUVECs and fibroblasts. A better model would be to incorporate actual human donor islets. The only image shown of this tissue is not of particularly high resolution and it is not possible to see whether the cells are healthy in the tissue. More importantly, the GSIS data for their control “islets” (static well) shows little insulin secretion in response to glucose stimulation. The authors report only 1.5x, whereas the vendor’s website suggests the secretion index should be higher (the company reports 10x). In panel (c) it seems that 2 out of the 9(?) experiments show very high stimulation in the GSIS assay, whereas many of the others do not look much different to

what is seen in, for example, “static with vasc.” or “static w/o vasc.” What accounts for this high variability? Also, why are the replicates seemingly so much fewer in the second, third and fourth groups? Are the symbols just on top of each other?

Finally, the authors note that this model can be used for immune cell perfusion, but don't sure any proof-of-concept experiments. Given they have used “islets” as a model organoid, presumably as a type I diabetes model, this would seem a worthwhile experiment to do.

Reviewer #3 (Remarks to the Author):

In this study, Quintard et al. present a novel microfluidic platform to trap and vascularize three-dimensional cell aggregates (e.g., spheroids and organoids). Using mesenchymal spheroids (made from HUVECs and fibroblasts), pancreatic spheroids as well as iPSC-derived blood vessel organoids, the authors confirmed a vascular connection and stable perfusability between the trapped organoids/spheroids and endothelial- and fibroblast-lined perfusion microchannels.

While focusing on mesenchymal and pancreatic spheroids as well as blood vessel organoids for proof-of-principle, with this innovative approach, Quintard et al. provide an original and versatile strategy that seems promising to be translated to vascularize other 3D tissues, such as other types of organoids.

Overall, it is an interesting and robust paper that provides solid results an in important area and provides a certain advancement over the state of the art.

The manuscript is very well written, the methodology used is sound and the results of the study are presented in a comprehensible way. The comprehensive supplementary material is very helpful for understanding the study in greater detail.

There are a couple of major and minor aspects that should be addressed:

General aspects:

- COC & on-chip oxygenation:

The microfluidic device is fabricated from COC, which is largely impermeable to oxygen. The authors should therefore address the impact of this oxygen impermeability throughout the manuscript, for example through simulations or on-chip oxygen sensing.

For example, in the flow conditions, does the oxygen perfused with the media flow meet the oxygen requirements of the integrated organoids/lined channels, or are they exposed hypoxia, especially when growing on the chip?

When comparing the flow condition to the static condition, it should be stated more clearly that the difference between these two conditions is not only the convective flow (and its associated shear), but also differences in nutrient- and oxygen concentrations, which the cells are exposed to.

- Permeability of established vascular networks:

The authors confirm the perfusability of the generated vascular networks between spheroid/organoid structures and endothelial-/fibroblast lining of the serpentine channels by perfusing 1 μm microbeads. However, since these beads are rather large, it would be interesting to see additional results on perfusion with smaller substances, such as a FITC dextran, for example, since this would also shed light on the vascular networks' permeability and barrier integrity. Moreover, the authors might further expand their EC markers by staining for tight junction markers.

- Versatility of trapping principle:

o The dimensions of the microfluidic trapping structures have to be adapted for different organoid sizes, and the success of the trapping depends on laminar flow regimes. It would be helpful if the authors calculated and outlined the lower and especially higher limits regarding organoid diameter, in which the trapping principle functions reliably. With 300 μm and 600 μm , respectively, the 3D aggregates injected in this study are rather small compared to other organoids, such as cerebral organoids, which easily reach diameters of 1-3 mm.

o The authors used a fibrin hydrogel for embedding and trapping the organoids, and lining the walls of the serpentine channels via the Landau-Levich-Bretherton effect. Would the same principle also work when using different hydrogel matrixes, which might have different polymerization- and rheological properties? For example, would the principle of the study also work when using the Matrigel-Collagen I mixture, used in original protocols of the BVOs?

- Limitations of the platform:

The manuscript would highly benefit from a short paragraph in the discussion part, which summarizes the current limitations of the system. This would give the reader a better idea of the model's window of functional stability and indications for expanding its context of use.

Specific comments to the manuscript:

- 101/530:

Polymerization of a fibrin hydrogel is very fast – is it possible to inject all 10 systems in parallel?

- 112: endothelialization of the serpentine channel

ECs and fibroblasts are encapsulated in the hydrogel, injected and pushed out (leaving only the walls covered) prior to polymerization. Does that imply that most ECs and fibroblasts are lost when the excess hydrogel is pushed out? This might be a major limitation when applying this principle for more valuable cell types than HUVECs, such as isogenic iPSC-derived vascular cells.

Also, would it not be possible to line walls and cover the hydrogel remnants by introducing an EC suspension after hydrogel polymerization and prior to connection of cell culture medium?

- Figure 3a:

o How is the quality of the EC barriers in the corner regions of the channel walls? They appear slightly discontinuous.

- 145:

Did the authors observe a disintegration of the networks after d13? Or, in other words, how was the limitation of 13 days defined?

- 274:

In supplemental video 5: What is the pulsatile movement of the tissue?

- 252 – 258/Figure 4c:

It would be very helpful if the authors could clarify this point: why are there no CD31+ cells inside the BVOs on the device, when the HUVECs around are missing, but there are CD31+ cells inside the BVOs in the controls in the U-bottom wells?

- 301/Figure 5: ECM remodeling

What is the evidence for “substantially enhanced” ECM remodeling under flow conditions? How/where is this depicted in figure 5?

- 369-370:

It is questionable whether all of the components in the cell culture media used to perfuse the vascular networks and organoids, such as FGF, would be stable over a two-week period in the incubator.

- 486:

Which medium was used for formation of the spheroids?

- 593:

Organoid extraction from the microfluidic platforms is a crucial aspect to allow for lysate-based readout methods. It is highly recommended to describe the extraction process in greater detail. Moreover, comments on robustness of organoid extraction from the devices would be valuable.

Point-by-point response to reviewers' comments

Color code:

- black = reviewers' comments
- bold dark blue = our comments
- light blue = initial text of our manuscript
- red = new manuscript text

REVIEWER COMMENTS

Reviewer #1 (Remarks to the Author):

In this paper, Quintard et al develop a microfluidic approach to vascularize organoids, and show its application in generating and perfusing HUVEC-based endothelial networks around mesenchymal and pancreatic islet spheroids, as well as in linking this HUVEC-based network with a vascular organoid. They show that these experiments remain viable up to 15 days in culture, and that some organoid characteristics are enhanced as a result of this perfusion. There have been multiple perfusion and vascularization platforms reported in the last few years, and this one interesting and novel in that it (1) repurposes a rather classic system of bead/single cell microfluidic trapping to trap larger sized organoids in a set of microfluidic traps and (2) combines this with the deposition of a endothelial cell-laden hydrogel, to efficiently (and presumably consistently, though this should be shown) coat the walls of the chip and crucially, a part of the organoid. Conceptually, this represents an elegant and new technical approach to the organoid vascularization problem, which would allow for robustness and scalability. As such, this platform represents a really promising technical advance for the field, and shows some interesting biological applications, particularly that gene expression of blood vessel organoids (BVOs) in flow and vascularized conditions on chip is closest to that of organoids transplanted in vivo, and that there are functional consequences to flow to flow with vascularization for glucose stimulated insulin secretion.

We are grateful to the reviewer for the insightful and positive feedback on the novelty and potential of our method.

There are nonetheless four important issues with this paper, which would have to be addressed: (1) Some of the novelty that the authors claim regarding their platform is overemphasized – indeed a number of systems exist which already vascularize organoids (including with some level of perfusion) – so the authors would have to be more clear on this point. “However, this conventional geometry does not allow reproduction of the fluxes observed in vivo, and no success on truly vascularized organoids has been reported yet.” What do the authors mean by truly vascularized? Penetration of flow the inner core of the organoid has already been shown in multiple systems.

We acknowledge that our terminology may have been confusing, our apologies. By “perfusion” we meant the convective luminal flow within the vasculature of organoids, and by “organoid” we denoted self-assembling 3D structures derived from stem cells.

It is worth noting that our wording was largely influenced by a 2021 review on vascularized organoids-on-chips (Zhang, S., Wan, Z. & D. Kamm, R. Vascularized organoids on a chip: strategies for engineering organoids with functional vasculature. *Lab. Chip* **21**, 473–488 (2021)): *“It should be noted that although various degrees of vascularization have been achieved in organoid models, none of these have yet been demonstrated to have fully perfusable function without in vivo transplantation. Thus, organoids cannot be considered as truly vascularized in vitro yet.”*.

In light of this review, and in light of our findings, we believe that the commentary from the review remains pertinent. We do, however, concede that the definition of “truly vascularized” can be ambiguous. To provide clarity, we have amended the manuscript accordingly: However, this conventional geometry does not allow reproduction of the fluxes observed in vivo, and replicating the in vivo functional vascularization of iPSC-derived organoids is still an ongoing challenge.

“Moreover, to date, no in vitro system has been convincingly shown to establish intravascular perfusion within organoids through functional anastomosis with neighbouring endothelial networks.” Again this has been done, for example: Integrating perfusable vascular networks with a three-dimensional tissue in a microfluidic device Nashimoto et al, 2017

This feedback aligns with the previous point, and our response is consistent. Nashimoto’s group’s significant contribution to the field utilized spheroids formed by aggregating HUVECs and fibroblasts (Nashimoto, Y. *et al.* Integrating perfusable vascular networks with a three-dimensional tissue in a microfluidic device. *Integr. Biol.* **9**, 506–518 (2017), **not iPSC-derived organoids as we reference. To ensure clarity and avoid potential misinterpretations, we have made appropriate revisions to our manuscript as described above. Additionally, based on the data from Nashimoto et al. discerning any distinct vasculature within the spheroid is challenging and from their presented images it is difficult to discern if the flow is merely surrounding the organoids or genuinely flowing through the internal vasculature.**

“With flexibility in application to a variety of bioengineering projects, this microfluidic platform supports the vascularization of nearly any 3D construct simply by adjusting perfusion channel and trap dimensions.” This statement should be toned down. Only three systems were tested, and success rates can differ between organoid systems.

We agree with the reviewer and have modified the text accordingly: With flexibility in application to a variety of bioengineering projects, this microfluidic platform supports the vascularization of various 3D constructs through an adjustable perfusion channel and trap dimensions.

Importantly, we have now added new findings which demonstrate the flexibility and suitability of our new method for epithelial organoids, e.g. lung organoids (detailed below). Additionally, we now provide an assessment of the efficiency of our trapping system (please see details below).

(2) One of the key advantages I see in this platform is the robustness, scalability and versatility of the platform – although other platforms may be able to achieve similar vascularization, this seems like it would be much faster and reliable - the authors should emphasize this point by showing and quantifying the variability of trapping efficiency, vascularization, both within a single chip as well as between chips.

We thank the reviewer for this positive feedback. We have now conducted additional experiments to quantitatively assess the trapping efficiency of our platform. The results of these experiments have been incorporated into the new Supplementary Fig. 4 (please see also data below). The trapping mechanism is consistently effective in basically every experiment we perform. On the rare occasions when the trapping fails, it is typically due to external technical issues not related to the chip itself, such as the organoid adhering to the pipette or the presence of debris in the microchannel. Please note that the microfluidic chips were thoroughly characterized prior to use, as indicated in Supplementary Fig. 1d.

We have added the following text: One of the strengths of our microfluidic platform is its robustness, scalability, and adaptability. Whereas other platforms might achieve comparable vascularization, the presented design promises enhanced speed and reliability. In practice, we can load a microchannel in just around 10 seconds, achieving a trapping efficiency that is close to 100% (Supplementary Fig. 4).

Supplementary Fig. 4. Organoid trapping and encapsulation efficiency. **a**, Exemplary images show organoids and polymer beads (ChromoSphere™) encapsulated in a fibrin hydrogel, utilizing our method across three distinct chip designs. **b**, Trapping yields at flow rates of 100, 300, and 500 $\mu\text{l}/\text{min}$ highlight consistent high trapping efficiency regardless of the flow rate. Trapping efficiency was assessed as number of successfully trapped organoids (or beads) per number of organoids (or beads) loaded on-chip ($n = 25$ organoids (or beads) per condition).

The text related to the organoid trapping methodology could be more clearly explained.

We apology if that was not clear, especially since we want to make this method accessible to the community. The Methods' text was modified accordingly:

Trapping mechanism and cell seeding procedure

The microfluidic setup is initiated using a syringe at the channel entrance, to sequentially introduce: (1) non-polymerized hydrogel embedding spheroids/organoids and endothelial cells, (2) air, and (3) growth medium. The channel features a serpentine loop that is bypassed by a U-cup shaped microchannel, which serves as a cell aggregate trap. When unoccupied, the hydraulic resistance of this trap (R_1 , Path A) is less than that of the serpentine loop (R_2 , Path B), thereby guiding spheroids/organoids into the trap by flow preference.

To initiate the seeding, a spheroid/organoid was gently extracted from a 96-well plate *via* a pipette tip and mixed into the hydrogel with thrombin. This mixture, amounting to 50 μ l and containing the spheroid/organoid, was placed into the reservoir leading to the microfluidic system. Using a syringe pump set to a withdrawal mode with a flow rate of $Q = 300 \mu\text{l}/\text{min}$, the spheroid/organoid progressed through the channel and was captured by the U-cup microchamber. Subsequent introduction of air served to position the hydrogel before it solidified. The Hydrogel remained at the U-cup, securing the spheroid/organoid and lining the channel corners. Both the hydrogel mix (prior to thrombin addition) and thrombin were kept at 4°C until used for the experiment. Immediately after mixing thrombin with the fibrin hydrogel, it was loaded onto the chip. This microchannel injection process takes approximately 10 seconds, largely excluding viscosity changes in the gel.

How is the trapping of the organoid occurring in fibrin which is presumably changing viscosity, and actually solidifying? Is the chip kept on ice or is there a limit of time in which this process can occur?

Indeed, the polymerization of the fibrin hydrogel occurs rapidly. We maintain the hydrogel mix (before thrombin addition) and the thrombin at 4°C until the time of use for an experiment. Upon adding thrombin to the fibrin hydrogel mix, we immediately load the gel onto the chip. Each microchannel injection is completed in approximately 10 seconds, ensuring that the change in the gel's viscosity during this brief period is negligible. We do not need to cool the chip on ice, being aware that other groups seed the hydrogel on ice. We now provide more details on this step in the Methods' section detailed above.

The efficiency of trapping should be quantified. How many organoids are typically trapped in a chip and does this vary between chips. One of the main advantages of this approach, as I see it, is the apparent robustness of the process, but this should be quantified and shown.

We agree that one advantage of our method is its robustness. As detailed above, we have now added quantitative data into our revised manuscript (please see Suppl. Fig. 4 above).

Additionally, it is unclear how many organoids were assessed in Fig. 3.

We acknowledge that we did not systematically indicate the numbers of organoids assessed. This oversight has been addressed in the revised manuscript. The exact sample size for each experimental condition is now indicated in the Figure legends. If not explicitly mentioned, all observations we have made across a large number of samples from many different experiments.

(3) Fibrin is used as the supportive hydrogel in all experiments, however fibrin used far less in organoids than Matrigel or synthetic gels. Is the trapping methodology also compatible with other gels? It would be important for the authors to show that this approach is more broadly applicable than just with fibrin, by that this approach is compatible at least with Matrigel, the standard gel used with organoids, and preferably with an additional gel system.

We employed fibrin in our experiments because HUVEC cells optimally self-organize in this matrix. Importantly, our methodology is not restricted to this gel type. Indeed, our approach is compatible with standard gels like Matrigel, or a Matrigel-Collagen I mixture, as used in the original protocol for generating BVOs. Although, in this study we did not aim at developing organoids on-chip, but rather to develop a reproducible and robust method to trap them, it is noteworthy that Matrigel polymerizes on-chip at a slower pace than a fibrin hydrogel, making our method even more applicable when using Matrigel. To underline this point, we now add new data using hiPSC-derived lung organoids in Matrigel (see new panel Fig. 1h below and in the revised paper). These organoids were not only efficiently trapped in our device but also exhibited robust growth and bud formation over a 10-day culture period on-chip.

We made the following changes to the text: In this study, our emphasis was on trapping vascular spheroids and organoids (generated off-chip) and culturing them in a fibrin hydrogel. However, this approach is versatile and applicable to other commonly utilized extracellular matrices (ECM) like Matrigel. For instance, by adhering to the same protocol, we cultured hiPSC-derived lung organoids within Matrigel (Fig. 1h). These organoids were not only efficiently trapped in the device but also exhibited robust growth and bud formation over a 2-week culture period on-chip.

Fig. 1. **h**, Representative images of vascular spheroid/organoid cultured in fibrin (left), and hiPSC-derived lung organoids cultured in Matrigel, showing efficient trapping and robust growth over 2 weeks on-chip (right).

Additionally, can the gel layer thickness around the organoid be varied? I.e. is the vascular network limited to the thin layer around the organoid?

Certainly, the gel layer thickness enveloping the organoid can be varied, but achieving this might be more complex. We dedicated substantial time examining this aspect, attempting to model the Landau-Levich-Bretherton phenomenon (the mechanism behind the residual thin layer of gel on the microchannels' walls post air bubble passage) within our chip setup. Given the unique design specifics of our configuration, the challenge is quite intricate. In the Landau-Levich-Bretherton framework, theory anticipates the gel layer thickness e on the wall to vary as $e = Ca^x$, where Ca represents the capillary number, and x varies from $\frac{1}{3}$ to $\frac{2}{3}$ depending on the particular experimental setup used (De Gennes et al. *Capillarity and Wetting Phenomena*. (Springer, 2004). doi:10.1007/978-0-387-21656-0. Kreutzer et al. *Multiphase monolith reactors: Chemical reaction engineering of segmented flow in microchannels*. Chem. Eng. Sci. **60**, 5895–5916 (2005)). **Through experiments conducted at diverse flow rates using Dextran FITC in our fibrin hydrogel, we ascertained the gel layer thickness e in the linear section of the microchannel.** The derived relationship $e \propto Ca^{0.31}$ aligns with documented findings in the existing literature. We have incorporated these results in the new Supplementary Fig. 3e.

The thickness of the gel layer on the microchannel walls can be adjusted by varying the flow rate of the air bubble passage (Supplementary Fig. 3e).

Supplementary Fig. 3e. Thickness e of the fibrin hydrogel layer following the air bubble passage at flow rate Q , relative to the capillary number Ca , in a straight section of the microchannel of width $2R$. The gel was visualized using Dextran FITC incorporated into the mixture (top view of the microchannel).

As for the thickness and configuration of the residual gel layer around the organoid within the trapping zone, it is a multifaceted issue. This is influenced by the selected flow rate and is affected by the precise trap design, organoid size and shape, organoid porosity, gel viscosity, and several other factors. Based on our empirical observations (see reviewer Fig. 1 below), a consistent pattern emerges: an increase in flow rate generally results in a thicker layer encasing the organoids. In the case of BVOs with HUVECs, a higher flow rate results in a more complete endothelialization of the serpentine channel from day 0. It is worth noting that experiments executed at significantly elevated flow rates (such as 10,000 μl/min, in contrast to our study's standard 300 μl/min) potentially impacts the efficacy of organoid trapping by producing air bubbles or by giving the organoid excessive inertia, causing it to escape from the trap.

Reviewer Fig. 1. Initial cell configuration at standard and high flow rates used for the introduction of the air bubble.

(4) Although the biological results are compelling, in some cases (BVOs) key controls are missing.

We acknowledge the oversight regarding the missing controls for BVOs. Indeed, these controls were conducted, but were inadvertently omitted from the first submitted manuscript amidst the vast amount of data. Here, we address each of the concerns in detail:

For example, “BVOs cultured on-chip in the fibrin hydrogel without HUVECs and fibroblasts show impaired growth, resulting in gel degradation, BVO’s shrinkage, and premature death (Supplementary Fig. 10).” Important control is missing: BVOs in fibrin in wells i.e. off-chip – to understand if it’s the fibrin that is inhibiting BVO growth.

When BVOs are cultured in fibrin within standard wells (off-chip), there is no observable difference when compared to those in grown in a Matrigel-Collagen I mixture used in the original protocol. The on-chip environment presents a distinct scenario due to the reduced gel volume, leading to observable gel degradation within a span of 24 to 48 hours. Such degradation can be observed in wells with both fibrin and Matrigel, but typically manifests after approximately one week. We added these data to Supplementary Fig. 10 to reflect these observations, stating: The growth and sprouting of these BVOs, when cultured in wells under identical conditions, remained unaffected because they were encapsulated in a substantial volume of gel, which protected against any gel degradation in the initial days of culture (see Supplementary Fig. 10b).

Supplementary Fig. 10b, Representative photographs of BVOs cultured alone in fibrin and Matrigel-Collagen I, exhibiting similar growth and sprouting abilities in both matrices.

“Fibroblasts, in comparison to HUVECs and hPSC-derived cell populations, produce more extracellular matrix (ECM), thereby preserving the gel structure. In the absence of fibroblasts, ECM production is limited, resulting in reduction in overall matrix and scaffolding support for the growth, development, and maturation of new vessel networks.” This doesn’t really make sense when one compares these results to very effective BVO growth off-chip in suspension without fibroblasts, but importantly without fibrin matrix.

We agree with the reviewer’s criticism and have removed this statement.

Additionally, BVOs include pericytes – can these be evidenced in these experiments, and do they have a role in maintaining the integrity of the vascular network?

Indeed, pericytes are present within the BVOs on-chip, as evidenced in Figure 3a in the manuscript and shown below. Their critical role in maintaining vascular structure integrity is well-documented. Although the presence of pericytes and smooth muscle cell coverage has previously been highlighted in our foundational publication on BVO generation (Wimmer, R. A. *et al.* Human blood vessel organoids as a model of diabetic vasculopathy. *Nature* 565, 505–510, (2019)), we opted to showcase only CD31 staining in our first submission of the current manuscript for simplicity. To rectify this oversight, we

have now incorporated pericyte staining in the new Figure 3b. This includes staining for pericytes (PDGFR \$\beta\$ ) and the basal membrane marker collagen (Col-IV) from our on-chip cultured BVOs, thereby presenting a more comprehensive view.

Similarly to BVOs cultured in wells, these BVOs self-organize on-chip into three-dimensional interconnected networks of *bona fide* capillaries containing an endothelial cell lined lumen, pericyte and smooth muscle cells coverage, and expression of tight junction proteins ZO-1 and adherens junction protein VE-cadherin (Fig. 3b). The organoids formed capillaries with hollow lumens surrounded by aprototypic basal membrane (Fig. 3c).

Fig. 3a, Schematic of the protocol used for the differentiation of human pluripotent stem cells into blood vessel organoids. **b**, Representative immunofluorescence of BVOs-on-chip, with capillary networks expressing CD31, the tight junction protein ZO-1, the adherens junction protein VE-cadherin, and covered by pericytes (PDGFR β). Experiments were repeated independently on $n = 6$ biological samples with similar results. **c**, Cross-section of BVO-on-chip showing capillaries with hollow lumens and collagen type IV basal membrane coverage. Experiments were repeated independently on $n = 10$ biological samples with similar results.

Additional comments:

-The ability to address each microchannel independently is a distinct advantage of the platform, which is clear in the text and supplementary figures but not so much from Figure 1. This could be emphasized graphically.

We agree and have accordingly added a new panel in Fig. 1c to show this.

Fig. 1c, Schematic diagram and photograph of the parallelization feature of our setup, showcasing 10 microchannels controlled simultaneously.

-It is mentioned that organoid size is 300-600um – the authors should show the range of organoid sizes which can be trapped in the current platform.

In the present study, our platform was designed to trap organoids of varied sizes. Specifically, we employed three different microchannel designs tailored for different spheroids/organoids: a cross section of 400 μm (suited for mesenchymal spheroids and pancreatic islet spheroids), 800 μm (for BVOs), and 1 mm (for lung organoids, a recent addition to our manuscript).

Regarding the upper size limit of organoids our platform can accommodate, we have not experimentally determined this yet. Nonetheless, based on basic theoretical assumptions, we are optimistic about extending this limit. Given the typical flow velocities ($v = 10$ mm/s) in our microchannels during chip loading, we can calculate the Reynolds number $Re = \frac{\rho v L}{\mu} = 6$, which remains low even in our largest microchannel design. To achieve a Reynolds number of 2000, marking the boundary for the laminar regime, our microchannels would need a characteristic length of $L = \frac{\mu Re}{\rho v} = 18$ cm. This indicates that there our design can accommodate even larger organoids (see below and new Supplementary Note 2 for the details of the calculation).

-Experiments are only carried out to D13-D14. What happens thereafter? The writing is ambiguous here: “network-like structures with a three-dimensional configuration developed and remained stable up to day 13 (Fig. 2d).”

Thank you for highlighting the ambiguity. We primarily chose day 13 for tissue fixation in our study as the HUVEC network is well-established by this time and remains stable, while the BVOs have had ample time to mature on-chip. While our study can be extended beyond this period, we refrained from including longer-term data in the initial submission to avoid presenting preliminary findings. We now conducted repeat experiments to confirm that our cultures can be maintained for up to 30 days. This result has been added in a new panel as Figure 4e.

Fig. 4e, Angiogenesis Analyzer outputs of endothelial networks on-chip assessing the total network length evolution over a period of 30 days. Experiments were conducted on $n = 4$ independent microchannels denoted as C_i (where i ranges from 1 to 4).

-“fluid velocities ranging from $v_{min} = 100 \mu\text{m/s}$ to $v_{max} = 7500 \mu\text{m/s}$ (Fig. 2i). Assuming laminar flow,” An approximation of the Reynolds number should be calculated and ascertained to laminar

Please see our calculation here, which has also been added to our revised manuscript:

With a calculated Reynolds number (Re) of 10^{-3} to 10^{-1} , the flow can be ascertained to be laminar (Supplementary Note 2); thus one can deduct from these values the fluidic shear rate at the vessel wall, which is given by $\dot{\gamma}_w = \frac{4\bar{v}}{R}$, where R is the radius of the vessel and \bar{v} the linear fluid velocity (Supplementary Note 3).

Supplementary Note 2: Reynolds number derivation (theory supplement)

The Reynolds number Re is given by: $Re = \frac{\rho v L}{\mu}$, where ρ is the density of the fluid, v is the flow speed, L is a characteristic length of the system and μ is the dynamic viscosity.

In our presented system, during the loading phase of the chip, we can choose $\rho \approx 1000 \text{ kg/m}^3$, $L \approx 1 \text{ mm}$, $v = \frac{Q}{S} \approx 5 \text{ mm/s}$ (using a flow rate of $Q = 300 \mu\text{l/min}$ in a microchannel of cross-section $S = 1 \times 1 \text{ mm}^2$), and $\mu \approx 0.9 \text{ mPa.s}$. This leads to a Reynolds number of $Re \approx 6$, ensuring a laminar regime.

During the long-term perfusion at $Q = 1 \mu\text{l/min}$, the fluid velocities in the vessels of radius $R = 15 \mu\text{m}$ were measured at $v_{min} = 100 \mu\text{m/s}$ and $v_{max} = 7500 \mu\text{m/s}$, thus corresponding to Reynolds number in the very low range of 10^{-3} to 10^{-1} .

These channels seem to be larger than in conventional microfluidics, so it is quite possible that turbulent flow may be occurring, which may potentially explain some recirculating regions seen in the bead experiments

As detailed above, despite of our microchannels being larger than in conventional microfluidics, we are still at very low Reynolds numbers.

-Why do the beads have preferential paths even outside the organoid, i.e. in the channels?

The beads are flowing along the path of least resistance. Because the microchannels are much larger than the typical vessel diameter, the beads preferentially flow in the channels.

There seem to be some high concentrations at some locations and it's not clear if these are zones of recirculation, blockage or something else. Can the authors explain these phenomena?

These zones (referenced in Fig. 2f) are artifacts resulting from our image processing method. The beads are not actually concentrated in the left and right corners, but they move slower in these regions. As detailed in the manuscript:

Of note, as the experiment was done at a constant flow rate, the velocity of a microbead was inversely proportional to the cross-sectional area it flowed through; hence the velocity decreased in regions without endothelium. This resulted in a prominent fluorescent signal from the beads' tracings on the bottom left and top right corners of the imaged area (Fig. 2f).

-Another missing (on-chip) control is: only BVO seeded without HUVEC cells to check if the microbeads cannot penetrate the BVO itself, i.e. whether the link with the HUVEC network is necessary for perfusion to occur.

It is a very important point as indeed the HUVEC network is necessary for perfusion of the trapped organoids. We have addressed this by including the relevant data in a new panel Supplementary Fig. 9c and added the following sentence to the manuscript:

In the absence of HUVEC endothelial networks, the BVO's vasculature was not perfused as assessed by microbeads perfusion, and no beads were detected in the trap site area (Supplementary Fig. 9c).

Supplementary Fig. 9c, Maximum of intensity projection over image stacks highlighting the tracks of the microbeads passing through the loop channel in the absence of HUVEC endothelial network.

-What genes were still differentially expressed between in vivo transplantation and in vitro on chip vascularization? It would be interesting to discuss this since it may suggest what is still needed to improve the system.

We now carried out a differential gene expression (DGE) analysis between vivo transplanted BVOs in mouse and the on-chip BVOs with flow and vascularization. The DGE analysis, using read count data per gene normalized by the RUVg method, identified 56 significantly differentially expressed genes ($p_{adj} < 0.05$, $|\log_2\text{FoldChange}| > 1$). Among these genes, 21 were induced and 35 were repressed in the in vivo matured BVO condition compared to the on-chip BVO condition with flow and vascularization. An ontological enrichment analysis carried out from these 56 differentially expressed genes did not identify any significantly enriched pathways. See Excel Reviewer File “DEG Pseudobulk VS Flow wt vasc.xlsx”.

It is worth noting, that 9 genes among the 16 genes up-regulated in the grafted organoids vs vascularized organoids-on-chip with a $\log_2\text{fold change} > 1.2$ (SNX10, CEACAM1, TMEM 176A, TMEM 176B, WNK4, CD1D, RSPO3, TNSF 15, MELTF) are involved in blood-related functions such as blood cell development (CEACAM1), blood vessel remodeling (RSPO3), positive regulation of plasminogen (MELTF), leukemia inhibitory factor (SNX10), and immune response (CD1D, TNSF15, CEACAM1). These results, although preliminary, suggest that the main differences between mice-engrafted organoids versus vascularized organoids-on-chip might be related to the presence of blood and the immune response in the former.

-The results of the glucose stimulated insulin secretion are compelling, however there is high variability in glucose response in the perfused system. Can the authors make a retrospective analysis of the vascular networks to determine if conditions where a “better” vascular network is associated to the enhanced insulin secretion?

This is a very interesting notion, however it was difficult to quantify/score differences in the endothelial network that could reflect change in the GSIS. Several previous reports have described temporally unstable oxygenation in vivo, particularly in the case of tumors, which have been described as “intermittent”, “acute”, “transient”, “cycling” hypoxia (Michiels et al. Cycling hypoxia: A key feature of the tumor microenvironment. *Biochim. Biophys. Acta BBA - Rev. Cancer* **1866, 76–86 (2016). Matsumoto et al.. Imaging Cycling Tumor Hypoxia. *Cancer Res.* **70**, 10019–10023 (2010)). We are likely facing the same issues that can explain the variability we observe in insulin secretion. To address this point, we propose to add the following sentence to the manuscript:**

The stimulation index for the pancreatic islet spheroids on-chip with both flow and vascularization showed higher variation, which might be attributed to differences in the quality or extent of the vascular network, and consequently in the possible occurrence of intermittent hypoxia.

Reviewer #2 (Remarks to the Author):

In this manuscript the authors present a microfluidic platform incorporating perfused vasculature and organoids. While the paper is well-written, much of the work is carefully performed, and the authors present some interesting methods, overall the work lacks significant novelty and the conclusions are somewhat overstated and not fully supported by the data.

We appreciate the reviewer's positive feedback on the quality of our work. In this response, we seek to further elucidate the novelty and significance of our research by adding new results and clarifying any misconceptions.

The authors claim "Importantly, we demonstrate effective anastomosis and controlled perfusion of the vascular organoids, as well as enhanced organoid growth, maturation, and vasculature development". Firstly, these "organoids" would better be described as spheroids as, with the exception of figure 6, they are not modeling any actual organs.

We apologize if the word organoid was used at the "wrong" place. We use the term "organoids" in alignment with the nomenclature within the stem cell biology community, where "organoids" refers to iPSC or ESC-derived 3D structures that somewhat resemble an organ. While it is true that in traditional anatomical terms, blood vessels are not classified as individual organs, in some contexts, especially in tissue engineering or regenerative medicine, "vascular organoids" or "blood vessel organoids" refer to lab-grown structures that mimic blood vessels. The iPSC-derived blood vessel organoids (BVOs) we employed in our study have been termed as "organoids" in the original Nature article from 2019 (Wimmer, R. A. *et al.* Human blood vessel organoids as a model of diabetic vasculopathy. *Nature* **565, 505–510 (2019). Wimmer *et al.*. Generation of blood vessel organoids from human pluripotent stem cells. *Nat. Protoc.* **14**, 3082–3100 (2019)), as well as in subsequent studies referencing this work. While BVOs are organoids, like the lung organoids we have added in this revised version, the two other models, that we have used, are clearly indicated as spheroids in the revised paper.**

Secondly, "growth and maturation" of organoids representing a specific tissue would be of considerable interest, however in this case the growth and maturation appears to refer to the formation of vascular networks – not a novel finding.

In essence, all iPSC-derived organoids rarely reach complete maturation and resemble more embryonic tissues rather than adult organs. Our BVOs will reach a higher degree of differentiation and maturity when grafted in mice (Nikolova, M. T. *et al.* Fate and state transitions during human blood vessel organoid development. 2022.03.23.485329 Preprint at <https://doi.org/10.1101/2022.03.23.485329> (2022)). In the present study, we report that we are able to observe on-chip and under flow condition, a maturation state that resembles the one obtained in vivo in mice. To the best of our knowledge, this is a key new finding.

Thirdly, a significant portion of the data show perfusion of beads through the organoids, demonstrating that HUVECs and fibroblasts cultured in a fibrin gel under flow generate perfusable networks. This has been demonstrated many times by several different groups and

again, is not at all novel. Other laboratories that have published substantially similar findings include those of Kamm, Chen, Hughes, Jeon, George, Huh, among others

The present manuscript takes a different approach to the previous research that the reviewer mentions. Of note, we of course cite these previous excellent studies in our manuscript (Whisler, J. A., Chen, M. B. & Kamm, R. D. Control of Perfusable Microvascular Network Morphology Using a Multiculture Microfluidic System. *Tissue Eng. Part C Methods* **20**, 543–552 (2014). Kim et al. Engineering of functional, perfusable 3D microvascular networks on a chip. *Lab. Chip* **13**, 1489–1500 (2013). Alonzo et al. Microfluidic device to control interstitial flow-mediated homotypic and heterotypic cellular communication. *Lab. Chip* **15**, 3521–3529 (2015). Sobrino, A. *et al.* 3D microtumors in vitro supported by perfused vascular networks. *Sci. Rep.* **6**, 1–11 (2016)). **By incorporating blood vessel organoids generated from human induced pluripotent stem cells into a serpentine microfluidic architecture, our approach offers multiple advantages over previously established microvascular network platforms. Using our unique design and readily translatable materials, we were able to grow endothelial networks on-chip, which not only arborized the organoids but, most importantly, also connected them functionally. Thus, this work provides a viable strategy to engineer fully functional microvasculature throughout the embedded organoids-on-chip. The device we have developed offers the flexibility to vascularize and perfuse other types of pre-endothelialized organoids, spheroids, tumoroids, or human tissue explants. To the best of our knowledge, this goes beyond the state-of-the-art. The novelty comes from 1) the microfluidic geometry and approach we used (all the groups mentioned above rely on the same general chip design); 2) we grow an endothelial network on-chip, like other groups, but using a completely new process and using this endothelial network as an “entrance”, a “connector”, rather than an end in itself; 3) we are able to connect successfully this endothelial (HUVEC) network to organoids and spheroids grown off-chip and subsequently trapped on-chip to be connected/anastomosed; and 4) to the best of our knowledge, we are the first to demonstrate on-chip organoid maturation resembling maturation observed in in vivo mouse-grafted organoids.**

In addition, there is a world of difference between perfusable vessels and tissue perfused exclusively through the vessels. The latter, obviously, represents the in vivo state but, many of these models have significant leak around (not out of) the vessels, which makes interpretation of their significance to the tissue very hard to determine. So, in this study, is the flow exclusively through the vessels?

We concur regarding the importance of achieving a system where perfusion occurs exclusively through the vessels to mirror the in vivo state. This remains an ongoing effort in our labs. While some groups have developed methods to achieve this (Wang, X. *et al.* Engineering anastomosis between living capillary networks and endothelial cell-lined microfluidic channels. *Lab. Chip* **16**, 282–290 (2016)), **this was not the primary focus of our current study. Depending on the specific configuration employed, there is likely some diffusion of the growth medium in the hydrogel inside the trap. Of note, for instance in the microchannel bead perfusion experiment shown in Fig. 2f and Supplementary Video 1, the arrangement of the cells was such that the flow was exclusively through the vessels.**

Do the vessels leak and is this leak at physiological levels?

The properties of HUVEC networks have already been extensively studied by other groups. Specifically, Hajal et al. demonstrated molecular permeability values that align with those anticipated in vivo (Hajal, C. *et al.* Engineered human blood–brain barrier microfluidic model for vascular permeability analyses. *Nat. Protoc.* **17**, 95–128 (2022)). They explored vascular permeability for molecules of different sizes, as illustrated in Fig. 13 of their publication. We have also observed similar results in our lab, with the HUVEC network auto-organized in a fibrin gel (see reviewer Fig. 2).

Reviewer Fig. 2. Dextran 10 kDa (red) in HUVEC vessels (green).

For the BVOs' vessels: This is indeed a pertinent question, yet, at this juncture, we do not have a definitive answer. It is worth noting that our system introduces a novel approach distinct from those documented in prior studies. While most microvascular-network chips consist of a thin layer of tissue (around 150 μm thick), our microchannels accommodate organoids up to 800 μm in size. This difference in scale makes conventional Dextran perfusion experiments challenging for our BVOs due to significant scattering. This limitation is also acknowledged in Hajal et al.'s article mentioned above, where they indicated that “*the signal is mostly lost at depths >50 μm in the gel because of scattering*” (Hajal, C. *et al.* Engineered human blood–brain barrier microfluidic model for vascular permeability analyses. *Nat. Protoc.* **17**, 95–128 (2022)). Additionally, the proximity of our microchannel walls to the organoids reflects the light from the Dextran flowing in the primary serpentine channel. Given these challenges, while it is essential to investigate the leakage of our BVOs cultured on-chip, a direct approach might not be immediately feasible. However, to provide insights into the vessel integrity, we have added data to show the presence of PDGFRβ+ pericytes, tight junctions (as defined by ZO-1 immunostaining) and the adherens junction protein VE-cadherin (new Fig. 3b), as well as collagen basal membrane around hollow capillaries (new Fig. 3c). We trust this adds valuable insights into the integrity of our vessels. Moreover, we reference our initial BVO study (Wimmer, R. A. *et al.* Human blood vessel organoids as a model of diabetic vasculopathy. *Nature* **565**, 505–510 (2019)) where “upon transplantation in mice, we did not detect any leakage from the BVO-derived in vivo human vessels post-Dextran perfusion”.

Fig. 3b, Representative immunofluorescence of BVOs-on-chip, with capillary networks expressing CD31, the tight junction protein ZO-1, the adherens junction protein VE-cadherin, and covered by pericytes (PDGFRβ). Experiments were repeated independently on n = 6 biological samples with similar results. **c**, Cross-section of BVO-on-chip showing capillaries with hollow lumens and collagen type IV basal membrane coverage. Experiments were repeated independently on n = 10 biological samples with similar results.

Also, with regard to the vessels, what diameter do they have and do they form hierarchical networks? Are they wrapped by pericytes, smooth muscle cells or both? These are critical components of organ vasculature and need to be determined for these “organoids”.

We thank the reviewer for pointing out this important issue. We indeed observed a hierarchical organization of the vessels, but we did not explore this in detail. Although the presence of pericytes and smooth muscle cell coverage was previously highlighted in the foundational publication on BVO generation, we opted to showcase only CD31 staining in our initial manuscript for simplicity. In retrospect, this decision seems to have omitted significant information. To rectify this oversight, we have now incorporated staining for pericytes (PDGFR β), smooth muscle cells (SM22), and collagen type IV from our on-chip cultured BVOs, thereby presenting a more comprehensive view. Please see Fig. 3b above. We also quantified the vessel diameters. This is now depicted in the new Figs. 5b and 5c, see below.

We added the following statements: When cultured under flow conditions, the vascularized BVOs displayed a physiological hierarchical organization with larger HUVEC vessels upstream and downstream of the BVO (Fig. 5b). The regions upstream and downstream with arteriole-venule sized vessels exhibited strong expression of the smooth muscle cell SM22 marker, while the narrow capillaries in the BVOs had hollow lumen structures surrounded by pericytes (Fig. 5b). The average diameter of the HUVEC vessels in the areas upstream and downstream of the BVO was measured to be 37 μm , corresponding to arterioles-venules sizes, whereas the vessels inside the BVO averaged 8 μm in diameter, aligning with in vivo capillary sizes (Fig. 5c).

b

c

Fig. 5b, Hierarchical structure of vascularized BVOs-on-chip, emulating the physiological arteriole-capillary-venule vascular tree, as schematically illustrated below (diagram created with BioRender.com). Experiments were repeated independently on $n = 6$ biological samples with similar results. **c**, Quantification of diameters for HUVEC vessels and BVO vessels ($n = 50$ individual vessels were measured for each type). Statistical significance was attributed to values of $P < 0.05$ as determined by unpaired t test (two-tailed). *** $P < 0.001$

The authors generated three organoid models: HUVECs and fibroblasts cultured for several days in a fibrin hydrogel; iPSC derived ECs cultured with fibroblasts in a fibrin hydrogel; and, a beta cell line cultured with fibroblasts and HUVECs in a fibrin hydrogel. All organoids were coated further in Fibrin+HUVECs+fibroblasts prior to being loaded into a custom microfluidic platform with a trap to catch the organoid. Following are some more specific concerns regarding the experiments presented:

Before addressing the specific concerns, we would like to provide some clarification. The spheroids or organoids used in this study were not coated in Fibrin+HUVEC+fibroblasts

prior to loading onto the chip. Instead, they were deposited within a hydrogel mixture (not polymerized yet) that included a single cell suspension of HUVECs and fibroblasts. Following the addition of thrombin to this mixture, which contained the spheroids/organoids, the solution was promptly loaded onto the chip as a liquid and will polymerize within a minute. As outlined in Figure 1 of our paper, this process resulted in the spheroids/organoids being encapsulated within a specific trap site on the chip, surrounded by the subsequently polymerized fibrin hydrogel.

Figure 1-3: the HUVECs+fibroblast model. The authors demonstrated that under flow, vasculature in their organoid connects with surrounding vessels. As noted above, dye perfusion of different molecular weights would be very helpful as an indicator of the “quality” of the vasculature and whether it is leaky.

Please see our comment above regarding dye perfusion experiments and leaky vasculature.

The authors also state that “The chip configuration described here resulted in anastomosis between a HUVEC endothelial bed and the capillaries of human blood vessel organoids. Consequently, our system can generate perfused hierarchical networks that encompass vessels ranging in size from arterioles to venules and capillaries.” Beside CD31 staining and network length, the authors don't investigate the hierarchical structure of their networks, nor do they examine endothelial artery/vein markers, nor do they demonstrate pericyte or SMC coverage, so it is hard to see how they can claim to have a hierarchical network comprised of arterioles/venules/capillaries.

We concur with the reviewer's observation that we did not provide direct evidence of a hierarchical network consisting of arterioles, venules, and capillaries. Our statement was primarily referring to the size range of the vessels, rather than their specific hierarchical classifications (“ranging in size from arterioles to venules and capillaries”). We recognize that this point was not adequately clarified. To address this concern, we performed new experiments and added new data to Figure 5 (please see Figure 5 above). We have also revised the related text in the manuscript accordingly, as detailed above.

Figure 3-4: iPSC-EC derived vascularized organoids encapsulated with HUVECs.

Again, we would like to take the opportunity to provide clarity on this point. The BVOs were not iPSC-EC derived; rather, they were generated using the protocol we initially described in Wimmer et al. Importantly, these BVOs are not composed solely of endothelial cells (EC). They also incorporate pericytes, mesenchymal stem-like cells, and hematopoietic cells. To showcase this, we added a new panel Figure 3a.

Fig. 3a, Schematic of the protocol used for the differentiation of human pluripotent stem cells into blood vessel organoids.

The authors demonstrated that when co-cultured with HUVECs under flow, their iPSC-EC derived vessels will lumenize and anastomose with HUVEC-derived vessels. An important control here would be BVO encapsulated with fibrin containing iPSC-ECs not HUVECs.

In this study, we were not aiming at growing an endothelial network on chip but instead we set out to develop a robust and flexible system that enables connection/perfusion of many types of 3D biological objects. The HUVEC network is only used as an “entry” to the BVO or any other type of organoids/spheroids. It is well known that HUVEC cells readily self-organize into perfusable tubes when embedded in a fibrin hydrogel. Our rationale was that the HUVEC network, which lines the microfluidic channel and trap, might be crucial for effective perfusion. This offers robustness and considerable flexibility in our system. Although exploring the replacement of HUVECs with iPSC-ECs is an intriguing suggestion, we hope that the referee accepts that such a control can be performed in future studies.

It is difficult to determine if the iPSC-ECs are maturing due to signaling from HUVECs or if the chip is playing a key role in iPSC-EC vessel maturation

To address this question, we now performed the experiments using 4 different conditions in parallel, namely 1) without HUVECs without flow, 2) with HUVECs without flow, 3) without HUVECs with flow, and 4) with HUVECs with flow. The data presented in Figure 6 shows that only the combination of flow and the presence of a HUVEC network enables maturation that is similar to the BVO grafted into mice.

In addition, although the authors say “Numerous BVOs vessels (CD31+GFP-) were observed near HUVEC networks, indicating functional perfusion from the HUVEC endothelial bed into the organoid vessels (Fig. 4b and Supplementary Fig. 7)”, there do not appear to be any data actually showing perfusion (beads, for example).

What we meant by this is that the anti-CD31 antibodies reached the BVOs’ vessels via functional anastomosis with the HUVEC network. We have now rephrased this sentence to avoid any misconception.

Numerous BVOs vessels (CD31⁺GFP⁻) were observed near HUVEC networks, **suggesting functional anastomosis** from the HUVEC endothelial bed into the organoid vessels (Supplementary Fig. 8a).

Importantly, to supplement and confirm these studies, we show the bead perfusion results in Fig. 5h-k of the revised manuscript.

Figure 5: This figure compares their BVO's covered in fibrin either with or w/o ECs & fibroblasts. The authors demonstrated that without including HUVECs and fibroblasts in the

fibrin coating, the BVO's begin to die and shrink. This is unsurprising given that the role of fibroblasts in supporting vasculature in fibrin gels is well documented.

We agree with the reviewer's criticism and have removed this statement.

The scRNAseq analysis seems to be very preliminary.

We apologize if that was unclear, but there is no scRNA-seq data in our study, only bulk RNA-seq.

The finding of matrix organization under flow is fully expected and not of particular interest.

Again, we would like to state that our microvascular-network chip design is new, and thus this is an important finding in our configuration – while it may not be in standard designs. Indeed, the constant and consequent ECM remodeling is necessary to maintain the structural integrity of the tissue within the trap site. Moreover, this finding aligns with our microscopy observations, thus it gives us confidence in our data. Finally, we think that studying the ECM organization is crucial due to the limited understanding we have about the signals and pathways that control the functional specialization of blood vessels, which includes insights from ECM-derived cues and signaling (Potente, M. & Mäkinen, T. Vascular heterogeneity and specialization in development and disease. *Nat. Rev. Mol. Cell Biol.* **18, 477–494 (2017)).**

The endothelial cells and stroma should be analyzed independently and cross-talk between he populations could be examined.

While we agree that this would be an interesting investigation, it would require scRNA-seq, a costly and time-consuming technique. In this consequent manuscript, we first aim to show a technology which represents a significant step forward over the state of the art. We hope that the community will find this platform useful, and will be able to conduct many different studies with it, including scRNA-seq profiling of various tissues cultured using our approach.

Are subpopulations of EC seen – for example, artery, vein and capillary?

This is an interesting point, but as we did not perform scRNA-seq, it is difficult to precisely answer this question. Moreover, the origins of arteriovenous patterning are widely debated. Coupled with the historic view that artery-venous lineage specification is a result of genetic pre-patterning prior to the onset of circulation, recent literature points to the mechanical relationship of flow parameters in arteriovenous patterning through high-low pressure gradients and shear stress modulation applied through everyday hemodynamic forces. The chip system presented here allows for high-dimensional analysis of arteriovenous patterning achieved through spatial transcriptomics and spatial sequencing technology not currently included in this manuscript. The anatomical orientation of the flow-derived vascular system makes it a prime candidate for high-resolution analysis of arteriovenous specification at all stages of vascular development, including exploration of the established genetic pre-patterning theories, and more recently, chemo-mechanically-

gated patterning achieved through angiogenic and vasculogenic vessel emergence and hemodynamic forces. Also, as detailed above, in the revised version of the manuscript, we better describe the hierarchical organization of the vessels we observed in our chips (Fig. 5b-c).

The comparison between the organoids in vitro and in vivo is an understandable choice, but of far more interest would be a comparison to endogenous vasculature/stroma in vivo.

A paper from Penninger's lab is in preparation on this very aspect. Our ambition here was to demonstrate that BVOs grown on vascularized chip grow and mature as well as when grafted in mice, the best maturation system we know to date.

Figure 6: The authors set out to demonstrate the actual utility of the device for supporting a pancreatic islet model, however this feels very under-developed. The authors generate their "islets" by culturing a commercially-available beta cell line with HUVECs and fibroblasts. A better model would be to incorporate actual human donor islets.

We appreciate the reviewer's suggestion regarding the incorporation of human donor islets into our device for a more physiologically relevant pancreatic islet model. We agree that using primary human islets would indeed provide a superior model for certain applications. Our current study, however, focuses on demonstrating the capabilities of our new microfluidic technology as a versatile platform for culturing 3D tissues, and, aside from the section on pancreatic islets, already comprises six substantial figures. The use of a commercially available beta cell line along with HUVECs and fibroblasts was intended as a proof of concept to showcase the potential utility of our device.

That being said, we have conducted preliminary experiments incorporating human islets from a single donor, and the results support the utility of our device for this purpose as well. These initial findings, albeit limited, are consistent with the outcomes using the beta cell line and indicate that our technology is suitable for use with primary human tissues (see below for details).

To utilize a more authentic model, we conducted further experiments using human pancreatic islets. These were dissociated and re-aggregated with endothelial cells and fibroblasts to create pre-endothelialized islets, which we termed Langerhanoids (reviewer Fig. 3a). Langerhanoids vascularized and cultured under flow as previously described showed a maintained response to insulin over 7 days (d+7), whereas native human islets in wells show a decrease of functionality at this stage (reviewer Fig. 3b). Interestingly, Langerhanoids vascularized on-chip showed an improved functionality at d+7 compared to human islets (reviewer Fig. 3c)). These data need to be further confirmed on other human islets donors, including the necessary controls, but due to donor scarcity it was impossible to perform so far.

Reviewer Fig. 3. Functionality assay of pre-vascularized native human islets (Langerhanoids) on-chip culture within a HUVEC endothelial bed and under flow conditions, compared to native human islets. **a**, Generation of pre-vascularized Langerhanoids, by using human islets which are then dissociated into single cells and reaggregated with FMA73 and RFP-HUVEC (diagram created with BioRender.com). **b-c**, Insulin secretions (**b**) and stimulation index = [High Glucose Solution]/[Low Glucose Solution] (**c**) at d+3 and d+7 of pre-vascularized native human islets (Langerhanoids) on-chip culture within a HUVEC endothelial bed and under flow conditions compared to native human islets and to Langerhanoids in-well. n=1 donor, each condition in triplicate. *To be further confirmed.*

The only image shown of this tissue is not of particularly high resolution and it is not possible to see whether the cells are healthy in the tissue.

We have changed the image in an updated panel Fig. 7a. Furthermore, the GSIS show that the cells are healthy enough to produce insulin in response to high glucose concentration, confirming that their functionality was not altered.

Fig. 7a, Generation of pre-vascularized pancreatic islet spheroids for on-chip culture within a HUVEC endothelial bed and under flow conditions (diagram created with BioRender.com).

More importantly, the GSIS data for their control “islets” (static well) shows little insulin secretion in response to glucose stimulation. The authors report only 1.5x, whereas the vendor’s website suggests the secretion index should be higher (the company reports 10x).

The values provided by the company concern cells grown in 2D. The company does not have values for these cells grow in 3D, and we are in close contact with them regarding this aspect as it is of high interest for them. In fact, our data show a reduced capacity of cells (or possibly delay of secretion capacity, or accessibility), but most importantly they are still secreting cells, in similar amount to native human islets.

In panel (c) it seems that 2 out of the 9(?) experiments show very high stimulation in the GSIS assay, whereas many of the others do not look much different to what is seen in, for example, “static with vasc.” or “static w/o vasc.” What accounts for this high variability?

First, we have updated the figure legend to clearly describe the exact number of samples used: We used biological replicates of n = 22, 5, 5, 5, and 11 per condition (wells, static w/o vasc., static with vasc., flow w/o vasc., and flow with vasc. respectively).

The inherent auto-organization of the network surrounding the islet could contribute to the observed variability in perfusion efficacy, which in turn may affect the outcomes of the GSIS assay. Several reports have described temporally unstable oxygenation in vivo, particularly in the case of tumors, which have been define by many terms such as “intermittent”, “acute”, “transient”, “cycling” hypoxia (Michiels, C., Tellier, C. & Feron, O. Cycling hypoxia: A key feature of the tumor microenvironment. *Biochim. Biophys. Acta BBA - Rev. Cancer* **1866**, 76–86 (2016). Matsumoto, S., Yasui, H., Mitchell, J. B. & Krishna, M. C. Imaging Cycling Tumor Hypoxia. *Cancer Res.* **70**, 10019–10023 (2010)). **We are likely facing the same kind of issues that can explain the variability we observe in insulin secretion.**

However, it is important to note that looking at the slopes in Fig. 7c may be misleading. To provide a more accurate evaluation, please refer to Fig. 7d, which shows the stimulation index: 8 out of the 11 stimulation index values from the “flow with vasc.” condition exceed all corresponding values from the other three chip conditions, underscoring a more consistent and pronounced response when dynamic flow and vascular components are present.

Also, why are the replicates seemingly so much fewer in the second, third and fourth groups? Are the symbols just on top of each other?

It was not possible to conduct all the conditions in parallel, as an internal control between our experiments, we performed the control condition in wells systematically as reference, and we emphasized the “flow with vasc.” condition to confirm the effect observed.

Finally, the authors note that this model can be used for immune cell perfusion, but don't sure any proof-of-concept experiments. Given they have used “islets” as a model organoid, presumably as a type I diabetes model, this would seem a worthwhile experiment to do.

We agree with the reviewer; this would seem a worthwhile issue to address in the future, but it would take way more than one experiments to thoroughly address it. Instead, we choose to delete this sentence.

Reviewer #3 (Remarks to the Author):

In this study, Quintard et al. present a novel microfluidic platform to trap and vascularize three-dimensional cell aggregates (e.g., spheroids and organoids). Using mesenchymal spheroids (made from HUVECs and fibroblasts), pancreatic spheroids as well as iPSC-derived blood vessel organoids, the authors confirmed a vascular connection and stable perfusability between the trapped organoids/spheroids and endothelial- and fibroblast-lined perfusion microchannels. While focusing on mesenchymal and pancreatic spheroids as well as blood vessel organoids for proof-of-principle, with this innovative approach, Quintard et al. provide an original and versatile strategy that seems promising to be translated to vascularize other 3D tissues, such as other types of organoids.

Overall, it is an interesting and robust paper that provides solid results in an important area and provides a certain advancement over the state of the art.

The manuscript is very well written, the methodology used is sound and the results of the study are presented in a comprehensible way. The comprehensive supplementary material is very helpful for understanding the study in greater detail.

We greatly appreciate the reviewer's positive and constructive feedback on our study. Your recognition of the innovative approach and its potential for broader applications reinforces the importance and relevance of our work.

There are a couple of major and minor aspects that should be addressed:

General aspects:

- COC & on-chip oxygenation:

The microfluidic device is fabricated from COC, which is largely impermeable to oxygen. The authors should therefore address the impact of this oxygen impermeability throughout the manuscript, for example through simulations or on-chip oxygen sensing.

Indeed, the choice to fabricate our chip in COC represents a distinct departure from many existing literature examples. As we have previously demonstrated with thorough on-chip oxygen sensing experiments in our lab, COC chips allow for precise control over the partial pressure of oxygen, unlike chips made from PDMS (Bussoo, A. *et al.* Real-time monitoring of oxygen levels within thermoplastic Organ-on-Chip devices. *Biosens. Bioelectron.* X **11, 100198 (2022)). While some researchers regard the gas porosity of PDMS as a benefit, our view is that its uncontrolled oxygen permeability can be just as problematic as its well-documented issue of drug binding.**

For example, in the flow conditions, does the oxygen perfused with the media flow meet the oxygen requirements of the integrated organoids/lined channels, or are they exposed hypoxia, especially when growing on the chip?

The oxygen requirements of the integrated organoids/lined channels may differ from organoids or spheroids. We did not precisely measure ppO₂ in each condition but we

assumed that, thanks to the flux, it was constant in the course of the experiments, unlike static conditions.

When comparing the flow condition to the static condition, it should be stated more clearly that the difference between these two conditions is not only the convective flow (and its associated shear), but also differences in nutrient- and oxygen concentrations, which the cells are exposed to.

Yes, the reviewer is right, we have modified the text accordingly: In comparing the flow condition to the static condition, it is essential to highlight that the distinctions are not limited to the convective flow and its associated shear; there are also increase in oxygenation and nutrient availability.

- Permeability of established vascular networks:

The authors confirm the perfusability of the generated vascular networks between spheroid/organoid structures and endothelial-/fibroblast lining of the serpentine channels by perfusing 1 μm microbeads. However, since these beads are rather large, it would be interesting to see additional results on perfusion with smaller substances, such as a FITC dextran, for example, since this would also shed light on the vascular networks' permeability and barrier integrity

The properties of HUVEC networks have already been extensively studied by other groups. Specifically, Hajal et al. demonstrated molecular permeability values that align with those anticipated in vivo (Hajal, C. *et al.* Engineered human blood–brain barrier microfluidic model for vascular permeability analyses. *Nat. Protoc.* 17, 95–128 (2022). They explored vascular permeability for molecules of different sizes, as illustrated in Fig. 13 of their publication. We have also observed similar results in our lab, with the HUVEC network auto-organized in a fibrin gel (see reviewer Fig. 2).

Reviewer Fig. 2. Dextran 10 kDa (red) in HUVEC vessels (green).

For the BVOs' vessels: This is indeed a pertinent question, yet, at this juncture, we do not have a definitive answer. It is worth noting that our system introduces a novel approach distinct from those documented in prior studies. While most microvascular-network chips consist of a thin layer of tissue (around 150 μm thick), our microchannels accommodate organoids up to 800 μm in size. This difference in scale makes conventional Dextran perfusion experiments challenging for our BVOs due to significant scattering. This limitation is also acknowledged in Hajal et al.'s article mentioned above, where they indicated that “the signal is mostly lost at depths $>50 \mu\text{m}$ in the gel because of scattering” (Hajal, C. *et al.* Engineered human blood–brain barrier microfluidic model for vascular permeability analyses. *Nat. Protoc.* 17, 95–128 (2022)). Additionally, the proximity of our microchannel walls to the organoids reflects the light from the Dextran flowing in the primary serpentine channel. Given these challenges, while it is essential to investigate the leakage of our BVOs cultured on-chip, a direct approach might not be immediately

feasible. However, to provide insights into the vessel integrity, we have added data to show the presence of PDGFR β + pericytes, tight junctions (as defined by ZO-1 immunostaining) and the adherens junction protein VE-cadherin (new Fig. 3b), as well as collagen basal membrane around hollow capillaries (new Fig. 3c). We trust this adds valuable insights into the integrity of our vessels. Moreover, we reference our initial BVO study (Wimmer, R. A. *et al.* Human blood vessel organoids as a model of diabetic vasculopathy. *Nature* 565, 505–510 (2019)) where “upon transplantation in mice, we did not detect any leakage from the BVO-derived in vivo human vessels post-Dextran perfusion”.

Moreover, the authors might further expand their EC markers by staining for tight junction markers.

Yes, to provide insights into the vessel integrity, we have incorporated data showcasing the presence of tight junctions ZO-1 and adherens junction protein VE-cadherin (new Fig. 3b), as well as collagen basal membrane around hollow capillaries (new Fig. 3c). We trust this will offer valuable insights into the integrity of our vessels.

Fig. 3b, Representative immunofluorescence of BVOs-on-chip, with capillary networks expressing CD31, the tight junction protein ZO-1, the adherens junction protein VE-cadherin, and covered by pericytes (PDGFR β). Experiments were repeated independently on n = 6 biological samples with similar results. **c**, Cross-section of BVO-on-chip showing capillaries with hollow lumens and collagen type IV basal membrane coverage. Experiments were repeated independently on n = 10 biological samples with similar results.

- Versatility of trapping principle:

The dimensions of the microfluidic trapping structures have to be adapted for different organoid sizes, and the success of the trapping depends on laminar flow regimes. It would be helpful if the authors calculated and outlined the lower and especially higher limits regarding organoid diameter, in which the trapping principle functions reliably. With 300 μm and 600 μm , respectively, the 3D aggregates injected in this study are rather small compared to other organoids, such as cerebral organoids, which easily reach diameters of 1-3 mm.

Previous publications have demonstrated that the design outlined in our paper can trap biological entities as small as individual cells. The primary challenge lies in creating microchannels with such small dimensions. For this purpose, materials like PDMS or silicone are required.

In the present study, our platform has been adapted to trap organoids of varied sizes. Specifically, we employed three different microchannel designs tailored for different organoids: a cross section of 400 μm (suited for mesenchymal spheroids and pancreatic islet spheroids), 800 μm (for BVOs), and 1 mm (for lung organoids, a recent addition to our manuscript – see also below for details).

Regarding the upper size limit of organoids our platform can accommodate, we have not experimentally determined this yet. Nonetheless, based on basic theoretical assumptions, we are optimistic about extending this limit. Given the typical flow velocities ($v = 10$ mm/s) in our microchannels during chip loading, we can calculate the Reynolds number $Re = \frac{\rho v L}{\mu} = 6$, which remains low even in our largest microchannel design. To achieve a Reynolds number of 2000, marking the boundary for the laminar regime, our microchannels would need a characteristic length of $L = \frac{\mu Re}{\rho v} = 18$ cm. This suggests there is room for accommodating even larger organoids (see below and new Supplementary Note 2 for the details of the calculation).

Supplementary Note 2: Reynolds number derivation (theory supplement)

The Reynolds number Re is given by: $Re = \frac{\rho v L}{\mu}$, where ρ is the density of the fluid, v is the flow speed, L is a characteristic length of the system and μ is the dynamic viscosity.

In our presented system, during the loading phase of the chip, we can choose $\rho \approx 1000$ kg/m³, $L \approx 1$ mm, $v = \frac{Q}{S} \approx 5$ mm/s (using a flow rate of $Q = 300$ μ l/min in a microchannel of cross-section $S = 1 \times 1$ mm²), and $\mu \approx 0.9$ mPa.s. This leads to a Reynolds number of $Re \approx 6$, ensuring a laminar regime.

During the long-term perfusion at $Q = 1$ μ l/min, the fluid velocities in the vessels of radius $R = 15$ μ m were measured at $v_{min} = 100$ μ m/s and $v_{max} = 7500$ μ m/s, thus corresponding to Reynolds number in the very low range of 10^{-3} to 10^{-1} .

The authors used a fibrin hydrogel for embedding and trapping the organoids, and lining the walls of the serpentine channels via the Landau-Levich-Bretherton effect. Would the same principle also work when using different hydrogel matrixes, which might have different polymerization- and rheological properties? For example, would the principle of the study also work when using the Matrigel-Collagen I mixture, used in original protocols of the BVOs?

We thank the reviewer for highlighting this aspect, as demonstrating the wide-ranging utility of our platform is of paramount importance to us. While we employed fibrin in our experiments because HUVEC cells have been observed to optimally self-organize in this matrix, our methodology is not restricted to this gel type. Indeed, our approach is compatible with standard gels like Matrigel, or a Matrigel-Collagen I mixture, as used in the original protocol for generating BVOs. It is noteworthy that Matrigel polymerizes at a slower pace than fibrin hydrogel, making our method even more straightforward when using Matrigel. To underline this point, we have incorporated findings from tests using hiPSC-derived lung organoids in Matrigel (see new panel Fig. 1h). These organoids were not only efficiently trapped in our device but also exhibited robust growth and bud formation over a 10-day culture period on-chip.

We made the following changes to the text: In this study, our emphasis was on trapping vascular spheroids and organoids (generated off-chip) and culturing them in a fibrin hydrogel. However, this approach is versatile and applicable to other commonly utilized extracellular matrices (ECM) like Matrigel. For instance, by adhering to the same protocol, we cultured

hiPSC-derived lung organoids within Matrigel (Fig. 1h). These organoids were not only efficiently trapped in the device but also exhibited robust growth and bud formation over a 2-week culture period on-chip.

Fig. 1. **h**, Representative images of vascular spheroid/organoid cultured in fibrin (left), and hiPSC-derived lung organoids cultured in Matrigel, showing efficient trapping and robust growth over 2 weeks on-chip (right).

- Limitations of the platform:

The manuscript would highly benefit from a short paragraph in the discussion part, which summarizes the current limitations of the system. This would give the reader a better idea of the model's window of functional stability and indications for expanding its context of use.

We agree with the reviewer on this comment, we have added a paragraph accordingly:

Several limitations of our platform merit discussion. First, the encapsulation process of ECs and fibroblasts in the hydrogel results in a significant cell loss when the excess gel is expelled before polymerization. This becomes particularly pronounced for larger designs intended for large organoids, where the gel volume, and consequently the number of cells required, escalates considerably. This poses challenges, especially when considering more valuable cell types like isogenic iPSC-derived vascular cells. Furthermore, while adjusting the gel layer thickness around the organoid is theoretically feasible, its practical execution is intricate. We invested significant effort into modeling the Landau-Levich-Bretherton phenomenon, which dictates the residual thin gel layer on the microchannels post air bubble passage. Our setup's unique design adds to this complexity, and addressing these challenges remains a priority as we refine the platform.

Specific comments to the manuscript:

- 101/530:

Polymerization of a fibrin hydrogel is very fast – is it possible to inject all 10 systems in parallel?

Yes, in theory absolutely. Currently, it is more convenient to perform one at a time to inject/control one by one, especially considering the solidification of the gel which is fast. However, we could imagine to pipet the 10 different mixtures, and inject them simultaneously using a 10-channel syringe pump.

- 112: endothelialization of the serpentine channel

ECs and fibroblasts are encapsulated in the hydrogel, injected and pushed out (leaving only the walls covered) prior to polymerization. Does that imply that most ECs and fibroblasts are lost

when the excess hydrogel is pushed out? This might be a major limitation when applying this principle for more valuable cell types than HUVECs, such as isogenic iPSC-derived vascular cells.

Indeed, the reviewer is right, it is a limitation of the method, and it is not clearly stated in our limitation paragraph. Let us note that we can easily reduce the volume of injection to 40 μ L of gel, thus there are not so many cells in total.

Also, would it not be possible to line walls and cover the hydrogel remnants by introducing an EC suspension after hydrogel polymerization and prior to connection of cell culture medium?

Yes indeed, it is possible, we have tried once as proof of concept (data not shown).

- Figure 3a:

How is the quality of the EC barriers in the corner regions of the channel walls? They appear slightly discontinuous.

We are not sure to understand but there is no discontinuity, please see Supplementary Fig. 3a.

- 145:

Did the authors observe a disintegration of the networks after d13? Or, in other words, how was the limitation of 13 days defined?

Thank you for highlighting the ambiguity. We primarily chose day 13 for fixation across this study as the HUVEC network is well-established by this time and remains stable, while the BVOs have had ample time to mature on-chip. While our study can be extended beyond this period, we refrained from including longer-term data in the initial submission to avoid presenting preliminary findings. However, we conducted repeat experiments to confirm that our cultures can be maintained for up to 30 days. This result has been added in a new panel Fig. 4e.

Fig. 4e, Angiogenesis Analyzer outputs of endothelial networks on-chip assessing the total network length evolution over a period of 30 days. Experiments were conducted on $n = 4$ independent microchannels denoted as C_i (where i ranges from 1 to 4).

- 274:

In supplemental video 5: What is the pulsatile movement of the tissue?

The pulsatile movement of the tissue is due to the pulses imposed with the syringe-pump to flow the beads. In order to maximize the chance of seeing beads perfusing the narrow vessels of the BVOs in a very limited amount of time (to avoid photobleaching), we had to use strong pulses, that led to tissue movements. It is now detailed in the Supplementary Video 5 legend:

Supplementary Video 5

Individual microbeads flowing through the blood vessel organoid's vasculature. Raw images (blue, green and red fluorescence). The two movies show microbeads (cyan) flowing within the blood vessel organoid's vasculature at two different areas and z-planes. **The observed pulsatile movement of the tissue is attributed to the strong pulses imposed by the syringe pump used to flow the beads, which was done to enhance the chances of observing beads perfuse the narrower vessels of the BVOs within a limited timeframe, minimizing photobleaching.**

- 252 – 258/Figure 4c:

It would be very helpful if the authors could clarify this point: why are there no CD31⁺-cells inside the BVOs on the device, when the HUVECs around are missing, but there are CD31⁺-cells inside the BVOs in the controls in the U-bottom wells?

Indeed, that is the purpose of the experiment. The staining through diffusion does not work on-chip because of the gel. In wells in media (not in the gel), the antibodies can diffuse and stain the BVO.

- 301/Figure 5: ECM remodeling

What is the evidence for “substantially enhanced” ECM remodeling under flow conditions? How/where is this depicted in figure 5?

Sorry for not being clear from the beginning regarding this point. The enhanced ECM remodeling under flow conditions is visible in Fig. 6a-b (or Fig. 5a-b in the previous version of the manuscript) on brightfield images at day 10. This is evidenced by the black structures around the trap site, corresponding to the cells' activity. These structures are absent under static conditions. We have clarified the text and the panel Fig. 6b.

Fig. 6a-b, BVO (brightfield, dotted cyan line) and HUVEC network (GFP) development from day 0 to day 10 on-chip, in static (a) and flow (b) conditions (diagrams created with BioRender.com). Enhanced ECM remodeling under flow conditions can be observed in the brightfield images at day 10, particularly by the presence of black structures surrounding the trap site, indicative of active cell processes (highlighted by white arrows).

- 369-370:

It is questionable whether all of the components in the cell culture media used to perfuse the vascular networks and organoids, such as FGF, would be stable over a two-week period in the incubator.

Indeed, the reviewer is right: we do not know. Both FGF-2 (Fibroblast Growth Factor-2) and VEGF-A (Vascular Endothelial Growth Factor-A) are sensitive growth factors that can degrade or lose activity over time, especially when exposed to elevated temperatures. This is why in practice, we fill up the syringe reservoirs everyday with fresh media, which by the way ensures a proper oxygenation as well.

This has been corrected accordingly in the manuscript to avoid any misunderstanding: To address concerns regarding the stability of sensitive growth factors in our cell culture media, such as FGF-2 and VEGF-A, we replenished the syringe reservoirs with fresh media daily.

- 486:

Which medium was used for formation of the spheroids?

We have added the information in the revised text: Fibroblasts and HUVEC cells were mixed at a ratio of 1:1 (5000 cells per well) in 150 μ l of medium consisting of a mix of CnT-ENDO / CnT-Prime Fibroblast medium (CELLnTEC, Bern, Switzerland) at a ratio 1:1.

- 593:

Organoid extraction from the microfluidic platforms is a crucial aspect to allow for lysate-based readout methods. It is highly recommended to describe the extraction process in greater detail. Moreover, comments on robustness of organoid extraction from the devices would be valuable.

We appreciate the comment of the reviewer and would like to take the opportunity to develop further on this point. Indeed, most existing microfluidic devices do not allow for tissue extraction – at least not easily. We made our device so that in practice, it is meant to be very easy. The chip is closed by a MicroAmp transparent adhesive film, which can be removed at any timepoint of the experiment in order to retrieve 3D biological objects. Of note, the tissues did not stick to the film, but remained in the microchannels. Subsequently, with a needle, the organoid was picked without the surrounding tissue, and immediately put in Trizol for lysis. We have now further detailed this in a new Supplementary Fig. 11.

Tissue extraction of the microfluidic chip

1 Removing of the MicroAmp adhesive film

2 Extraction of the organoid using a needle

3 No surrounding tissue is collected

4 Lysis of the organoid in Trizol

Supplementary Fig. 11. Tissue extraction. Organoid extraction process from the microfluidic device. The microfluidic chip is sealed with a MicroAmp transparent adhesive film, designed for easy removal at any desired timepoint during the experiment. Upon removal of the adhesive film, 3D biological tissues remain positioned within the microchannels. For extraction, a needle is carefully used to pick the organoid without disturbing surrounding tissues, after which it is promptly placed in Trizol for immediate lysis. This design ensures ease and robustness in the organoid retrieval process from the microfluidic platform.

REVIEWERS' COMMENTS

Reviewer #1 (Remarks to the Author):

The authors have addressed all my comments in significant and thoughtful detail. They have made substantial changes to the manuscript as well as performed the required additional key experiments. I believe the manuscript makes a substantial contribution to the organoid vascularization field as it is now, and I would advise its publication.

Reviewer #2 (Remarks to the Author):

The authors have added a considerable amount of new data and explanation, and many points have now been greatly clarified. They are commended for their efforts.

A key problem remains that will severely limit the utility of the platform for many studies, and that is the problem of whether perfusion of medium is exclusively through vessels (as in vivo) or whether some (much) spills around the outside. In the latter case the vessels become somewhat (entirely) superfluous. At best, it makes studies on delivery of drugs to tissues very hard to interpret. It is good that the tissues “behave” once transplanted into mice, but having them function in vitro as they do in vivo would be hugely useful. From the authors description it appears that this issue has not yet been solved. Clearly there is still important work to be done here.

That said, the design is a useful advance on the way to fully functional perfused tissues in vitro.

Reviewer #2 (Remarks on code availability):

Not competent to review code.

Reviewer #3 (Remarks to the Author):

All my concerns and questions in the initial version of this study have been addressed far beyond expectations with this revised manuscript! Quintard et al. have meticulously and comprehensively approached the reviewers' comments.

The authors have strengthened and clarified the manuscript by significantly expanding experimental investigations and complementing results to all aspects which had previously been lacking clarifications. I was especially excited to see that the authors have included (i) another organoid system as well as other hydrogel matrices to further highlight the versatility of the developed technology and (ii) data on the robustness of the on-chip vascularised organoids in long-term experiments.